



# Improved calibration procedures for the EM27/SUN spectrometers of the COllaborative Carbon Column Observing Network (COCCON)

Carlos Alberti[1], Frank Hase[1], Matthias Frey[2], Darko Dubravica[1], Thomas Blumenstock[1], Angelika Dehn[3], Gregor Surawicz[4], Roland Harig[4], Johannes Orphal[1,a] and the EM27/SUN-partners team[5]

[1]Institute of Meteorology and Climate Research (IMK-ASF), Karlsruhe Institute of Technology (KIT), Karlsruhe, Germany
[2]National Institute for Environmental Studies (NIES), Tsukuba, Japan
[3]ESA / ESRIN, Frascati, Italy
[4]Bruker Optics GmbH, Ettlingen, Germany
[5]See section: Team list
[a]Now at: Division 4 "Natural and Built Environment", Karlsruhe Institute of Technology (KIT).

*Correspondence to*: Carlos Alberti (carlos.alberti@kit.edu)

*Abstract.* In this study, an extension on the previously reported status of the COllaborative Carbon Column Observing Network's (COCCON) calibration procedures incorporating refined methods is presented. COCCON is a global network of portable Bruker EM27/SUN FTIR spectrometers for deriving column-averaged atmospheric abundances of greenhouse gases.

The original laboratory open-path lamp measurements for deriving the instrumental line shape (ILS) of the spectrometer from water vapour lines have been refined and extended to the secondary detector channel incorporated in the EM27/SUN spectrometer for detection of carbon monoxide (CO). The refinements encompass improved spectroscopic line lists for the relevant water lines and a revision of the laboratory pressure measurements used for the analysis of the spectra. The new results are found to be in good agreement with those reported by Frey et al. (2019), and discussed in detail. In addition, a new

calibration cell for ILS measurements was designed, constructed and put into service. Spectrometers calibrated since January 2020 were tested using both methods for ILS characterisation, open path (OP) and cell measurements. We demonstrate that both methods can detect the small variations of ILS characteristics between different spectrometers, but the results of the cell method indicate a systematic bias of the OP method. Finally, a revision and extension of the COCCON network instrument-to-instrument calibration factors for $XCO_2$, XCO, and $XCH_4$ is presented, incorporating 47 new spectrometers (of 83 in total

by now). This calibration is based on the reference EM27/SUN spectrometer operated by the Karlsruhe Institute of Technology (KIT) and spectra collected by the collocated TCCON station Karlsruhe. Variations in the instrumental characteristics of the reference EM27/SUN during 2014 to 2017 were detected probably arising from realignment and the dual-channel upgrade performed in early 2018. These variations are considered in the evaluation of the instrument-specific calibration factors in order to keep all tabulated calibration results consistent.

## Introduction

The activities of modern mankind have detectable negative impacts on the atmosphere, including the release of various trace substances into it, mainly due to industrialization, to the globalization of the economy and related transport, and to increasing





power generation and land use. Among them, greenhouse gases (GHGs) directly affect the Earth's radiative balance because
they reduce the thermal infrared emission to space. Due to their long lifetime, those gases affect the climate for decades or
centuries. The reduction of GHG emissions is thus recognized as important and urgent political and societal task. Although
there are several species categorized as GHGs, the gases mainly responsible for the increasing global warming are carbon
dioxide ($CO_2$) and methane ($CH_4$). The anthropogenic emissions of $CO_2$ from fossil fuel combustion are the main driver of
global warming which will reach and most likely exceed 1.5° C within the next coming two decades (IPCC AR6 WG1, The
Physical Science Basis, 2021). Methods for monitoring and quantifying those gases - thereby pinning down their sources and
sinks as well as their links with various human activities - are essential for appropriate decision-making to mitigate climate
change. In general, atmospheric concentrations of GHGs are categorized into in-situ and remote sensing techniques; the first
of these techniques offers very high accuracy but faces problems due to its high sensitivity to emitting sources in the vicinity
and to details of the vertical transport. Ground-based remote sensing techniques using solar absorption spectroscopy deliver
column-averaged atmospheric GHG abundances, but suffer from lower precision and accuracy and lower sensitivity for local
sources. These observations are, however, well suited for the validation of satellite missions and observations of larger source
regions. Currently, several dedicated GHG satellite missions are in orbit: the Greenhouse Gases Observing Satellite (GOSAT)
and GOSAT-2 (Kuze et al., 2009; Morino et al., 2011; Yoshida et al., 2013; Suto et al., 2021), the Orbiting Carbon
Observatory-2 (OCO-2) and OCO-3 (Frankenberg et al., 2015; Crisp et al., 2017; Eldering et al., 2017 and 2019), the
Copernicus Sentinel-5 Precursor (S5P) (Veefkind et al., 2012), and the Chinese Carbon Dioxide Observation Satellite (TanSat)
(Liu et al., 2018). The ground- based Total Carbon Column Observing Network (TCCON) (Wunch et al., 2011) is the most
important source for reference validation data and has been recently supplemented by the portable EM27/SUN spectrometers
managed by the COllaborative Carbon Column Observing Network (COCCON, Frey et al., 2019). Both networks use solar-
viewing Fourier Transform Infrared (FTIR) spectrometers. There, column-averaged atmospheric abundances of GHGs are
derived from the observed near-infrared spectra.

The TCCON was established in 2004 to obtain accurate measurements of column-averaged dry-air mole fractions of $CO_2$, CO,
$CH_4$ and $N_2O$ (denoted as Xgas, e.g. $XCO_2$). The TCCON stations operate high resolution Fourier transform IFS125HR
spectrometers (FTS) manufactured by Bruker. Nowadays the network has 29 operational sites worldwide (Wunch et al., 2011).
Although these TCCON stations are distributed around the globe, there are still considerable geographic gaps lacking
measurements. The COCCON network emerged in 2016, based on the low-resolution (0.5 $cm^{-1}$) EM27/SUN FTIR
spectrometer developed by KIT in cooperation with Bruker (Gisi et al., 2012; Hase et al., 2016) which delivers similar precision
and accuracy as TCCON, assuming a careful calibration of each spectrometer. Several studies have revealed its previously
unprecedented high level of performance and stability (Frey et al., 2015, 2019; Sha et al., 2020). In addition, the portability of
the EM27/SUN spectrometer favours campaign use, and a series of such campaigns have already successfully been conducted
by various investigators (Hase et al., 2015; Hedelius et al., 2016; Butz et al., 2017; Viatte et al., 2017; Chen et al., 2016, 2020;
Kille et al., 2019; Vogel et al., 2019; Luther et al., 2019; Makarova et al., 2021; Dietrich et al. 2021 and Jones et al. 2021).



As the number of EM27/SUN spectrometers being deployed across the world continues to grow, it is essential in order to maintain and further improve the reliability of the network to keep up with proper quality assurance/quality control (QC/QA) work and to apply the best available procedures for instrumental characterisation on new units before they go into operation.

In this regard one of the most important parameters, which needs to be specified for a defined FTIR spectrometer prior to the analysis of its atmospheric measurements is the Instrumental Line Shape (ILS). Several studies have shown that the real ILS deviates from the ideal one (e.g. Hase et al., 1999; Bernardo et al., 2005), e.g. due to interferometric misalignment, optical aberrations, or uneven illumination or sensitivity of the detector element. An out-of-range ILS result points to instrumental issues which need to be alleviated by realignment or replacement of optical or mechanical components. A good knowledge of

the near-nominal ILS is imperative for a precise GHGs retrieval. Portable spectrometers need to demonstrate their ability to preserve their ILS characteristics during transport events and over sufficiently long periods of time. Stable instrumental characteristics have been demonstrated despite harsh transport and conditions of operation for periods of up to several years (Frey et al., 2015.

In this paper, the open path (OP) method for ILS calculation of EM27/SUN spectrometers as described by (Frey et al., 2015) is significantly improved and applied to further spectrometers. Additionally, a new calibration cell filled with $C_2H_2$ was designed, built and used in addition to the open-path method since January 2020. The OP improvements are described in sub-section 1.2; the cell method is described in section 2. This provides additional redundancy of the ILS characterisation and allows the comparison of both approaches, OP and cell measurement. The results achieved with the OP and cell methods are

described and compared in sections 3 and 4, respectively. We confirm that both methods have the ability to detect residual instrument-specific deviations from the nominal ILS. The cell method suggests that there is a bias in the open path results, probably generated by incorrect pressure broadening parameters of the relevant $H_2O$ lines. Section 5 is devoted to the discussion of the solar side-by-side measurements. In section 5.1, we present the continued long-term trace gas measurements of the reference EM27/SUN spectrometer used as fixed point for the instrument-specific gas calibration factors of the other

EM27/SUN units. This time series now spans seven years. We compare these atmospheric measurements with the trace gas amounts derived from low-resolution spectra collected with the co-located 125HR spectrometer of the Karlsruhe TCCON station. Variations of the reference unit's instrumental characteristics connected to realignment and the dual channel upgrade performed in early 2018 have been identified. In section 5.2 we show the results from the re-evaluation of the open-path measurements and list the instrument-specific calibration factors for each spectrometer, considering the detected variations of

the reference unit. The resulting survey of instrumental characteristics is a considerable extension of the work by Frey et al. (2019), as it contains results for 47 new spectrometers. In section 5.3, the spectral signal-to-noise characteristics for all investigated spectrometers is summarized.



## 1 Advancing the open-path method for ILS characterisation

The method described in Frey et al. (2015), of characterising the ILS of low-resolution spectrometers using open path
measurements is improved and extended; the method is briefly summarized here along with a description of the main
improvements. The idea of the approach is to use the absorption of infrared radiation from an external tungsten lamp by strong
water vapour lines along a laboratory path of a few meters. A fit to the spectrum is performed by adjusting the $H_2O$ column, a
spectral scaling factor, a spectrally variable background transmission level, and a parameterized ILS. Two parameters are used
for describing deviations from the nominal ILS shape: the "modulation efficiency amplitude" (MEA) describes a deviation
from the expected ILS width, the "phase error" (PE) quantifies the asymmetry of the ILS (Hase et al., 1999). Because the
widths of the spectral lines generated along the open path depend on pressure and temperature, these parameters need to be
recorded for the analysis of the measurements. The self-broadening of $H_2O$ is a non-negligible contribution; therefore, the
absorption path length needs to be known. The $H_2O$ partial pressure is calculated from the retrieved $H_2O$ column amount,
pressure and temperature, so the analysis of the spectrum is an iterative procedure (repeated until convergence to a self-
consistent solution is reached).

### 1.1 Procedure and setup

The general setup is described by Gisi et al. (2012) and Frey et al. (2015) and illustrated in Figure 1. At least two hours before
the first interferograms are collected, the spectrometer is powered up. Two openings in the spectrometer's shelter are uncovered
for allowing exchange between the air trapped inside the spectrometer and the external laboratory air. This equilibrates the
water vapour mixing ratio inside the spectrometer with the environment and allows the spectrometer to reach a stable operating
temperature, thereby minimizing spectral drifts of the He-Ne laser which controls the interferogram sampling. For the radiation
source, a commercial halogen lamp attached to a lens collimator is used. The lamp bulb is grounded on the outside and is tilted
with respect to the optical axis to minimize channelling artefacts (Blumenstock et al., 2021). The spectrometer resides on a
table, while the lamp is mounted on a tripod at about 4 m distance from the first mirror of the solar tracker attached to the
spectrometer. The position of the lamp is level with the tracker and the beam is steered towards the first tracker mirror.





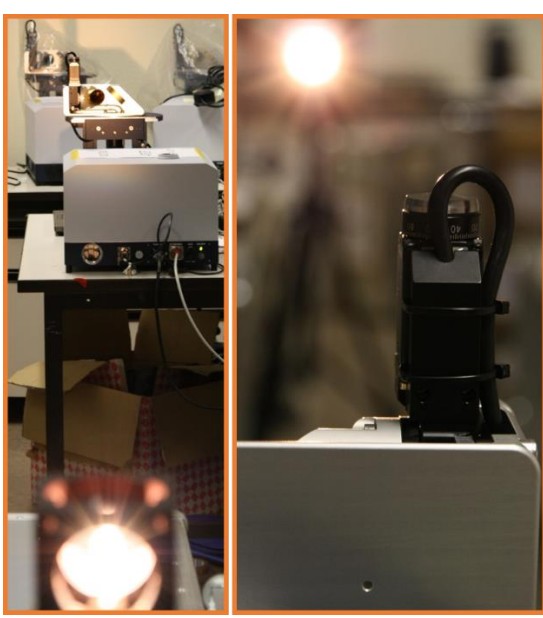

**Figure 1: Set-up of the open-path measurements. The left panel shows a view from the lamp unit (bottom) towards the spectrometer with its solar tracker (top): the lamp in the bottom part and the instrument located in a distance of 4.20 m. The right panel shows the view in the opposite direction from the spectrometer towards the lamp unit.**


## 1.2    Updated measurement procedures

The main changes with respect to the old method are described in the following sub-sections.

### 1.2.1    Geometry of the set-up:

 in addition to the open-path procedure (January 2020), a cell setup has now been implemented for the calibration of the
EM27/SUN spectrometers; the geometrical arrangement previously used was slightly reconfigured to support both OP and cell measurements. The spectrometer is now oriented in such a way that the cell can be conveniently located in the infrared beam on top of the spectrometer housing (see Figure 1 and Figure 4 A)). This modification results in a slightly larger distance between the lamp and the first tracking mirror; in the past that distance was 4.0 m and it is now 4.2 m.

### 1.2.2    Distance travelled by the beam inside the instrument

We decided to re-check and thereby noticed that the previously assumed optical path length inside the spectrometer was underestimated. In order to derive this distance properly, an optical method was applied. It uses a digital camera, a finely structured optical target printed on a piece of paper and pocket lamp for illumination of the target. Figure 2 demonstrates the method. The aim is to optically measure the inaccessible path section from the instrument entrance window until position E), as shown in that figure.






For performing the distance measurement, the solar tracker was unmounted to gain access to the entrance window. The paper target was located at position E) and illuminated with the pocket lamp. The digital camera equipped with a telescopic lens was positioned directly in front of the entrance window for observing the target. The target is focused properly and the focus position of the lens is maintained while the spectrometer is removed. Next, the target is arranged at such a distance from the

lens that a sharp image is re-created. This distance can easily be measured geometrically, we estimate the accuracy of the method to be better than 5 cm. In order to determine the complete optical path inside the instrument, the distance E) to F) and F) to I) are measured with a conventional ruler and added to the distance calculated with the previously explained method.

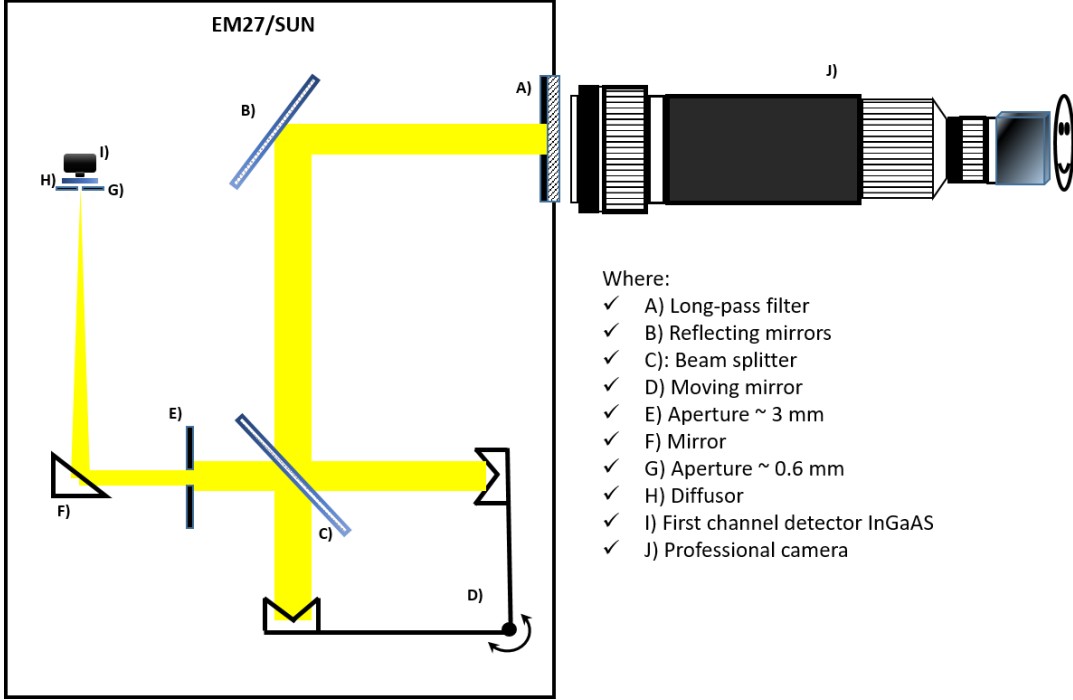

**Figure 2: Light path of the beam inside the instrument coming from the tracking mirrors.**

In Table 1, the old and new results for the relevant distances are presented. Note that the distance between lamp and first tracker mirror has been changed deliberately.  The corrected other two contributions to the total path length, which are used for the proper calculation of the $H_2O$ partial pressure, have been considered in the reanalysis of the old lamp measurements.

For the analysis of the lamp measurements after mid-January 2020, the updated values as provided in Table 1 have been used. The effect on the ILS parameters via the resulting change in $H_2O$ partial pressure is small, but detectable. We discuss this effect in section 3.1.





**Table 1: Description of the main changes in the path distance used in the past and the current ones.**

| Length | Old | New (cm) | Difference abs(new-old)/old |
|---|---|---|---|
| Lamp to 1st tracking mirror | 400.0 | 430.0 | 7.5% (note: deliberate adjustment) |
| First tracking mirror to Spectrometer housing entrance | 38.0 | 33.0 | -13.2% |
| Spectrometer housing entrance to detector | 58.0 | 74.0 | +27.6% |


### 1.2.3    Measurement procedure

Before the collection of measurements, the tracking mirrors (elevation and azimuth) are carefully adjusted in order to centre the lamp image on the field stop. The image of the lamp needs to surpass the field-stop's diameter. This procedure is conveniently carried out using the camera, which is incorporated in the spectrometer for controlling the solar tracking.

### 1.3    Data-acquisition and improved processing

Before the interferograms are recorded (either with or without the cell in the path), the pre-gain and gain settings of both detectors are checked. The manufacturer's data acquisition software OPUS is used to perform the measurements and to process the DC-coupled interferograms. Ten double-sided full resolution scans recorded with 10 kHz scan speed are co-added into one averaged interferogram, thirty to forty averaged interferograms of this kind are recorded in total to achieve a spectral signal-
to-noise ratio in the range of 2000 to 3000, see Section 5.3. As the DC level of the EM27/SUN is slightly variable as function of optical path difference, a DC correction is applied (because the solar observations also undergo a DC correction). The resulting spectra are normalized to about unity in the spectral range required for the ILS analysis and are stored with a zero-filling factor of eight to support the visual inspection of the spectral fit quality.

### 1.3.1    Required auxiliary data

In order to correctly derive the $H_2O$ column and ILS parameters, pressure and temperature need to be measured, both inside the instrument and outside in the laboratory. The temperature inside the spectrometer is recorded using the sensor built into the spectrometer by the manufacturer. The temperature of the laboratory air is recorded using digital thermometers offering 0.8° C accuracy (Lutron MHB-382SD data loggers or Thermo Hygrometer Barometers of type PCE-THB 40 were used).





While in the previous data analysis, the pressure readings of the aforementioned portable sensors were used, we migrated for the analysis of new measurements as well as for the re-analysis of previous measurements to the pressure record from the nearby meteorological tall tower. This tower is operated by the Institute for Meteorology and Climate Research - Department Troposphere (IMK-TRO), see Kohler, et al. (2017). The pressure sensor used at the tower is calibrated in regular intervals and the data accuracy is expected to be within 0.5 hPa. The tower is located at a distance of about 800 m to our laboratory. We

apply a barometric correction to the pressure data measured at the tower, as the elevation of the laboratory is higher than the location of the pressure sensor by ca. 11 m.

### 1.3.2    Data processing

   For the retrieval of the ILS parameters, the LINEFIT software version 14.8 (Hase et al., 1999) is used. In order to retrieve the

$H_2O$ column, a simple two parameter ILS model is utilized as described in (Frey, et al., 2015). The main extension in the retrieval setup is that the ILS is now characterized for both the primary and the CO channel. Two different spectral regions are therefore investigated as shown in Figure 3. The previously used spectral window covers 7000-7400 cm$^{-1}$, the newly added window covers 5275 - 5400 cm$^{-1}$. The latter window resides in the spectral overlap region covered by both detectors, allowing a check for a degraded ILS of the CO channel with respect to the primary channel, because in this spectral window the retrieval

of ILS parameters can be performed from both main channel and CO channel spectra. By comparing the ILS parameters retrieved from the same spectral window, biases introduced by spectroscopic inconsistencies cancel out. So, according to the new scheme presented here, three sets of ILS parameters are retrieved and the two additional retrievals performed in the new window are introduced for recognizing a potential misalignment of the CO detector.



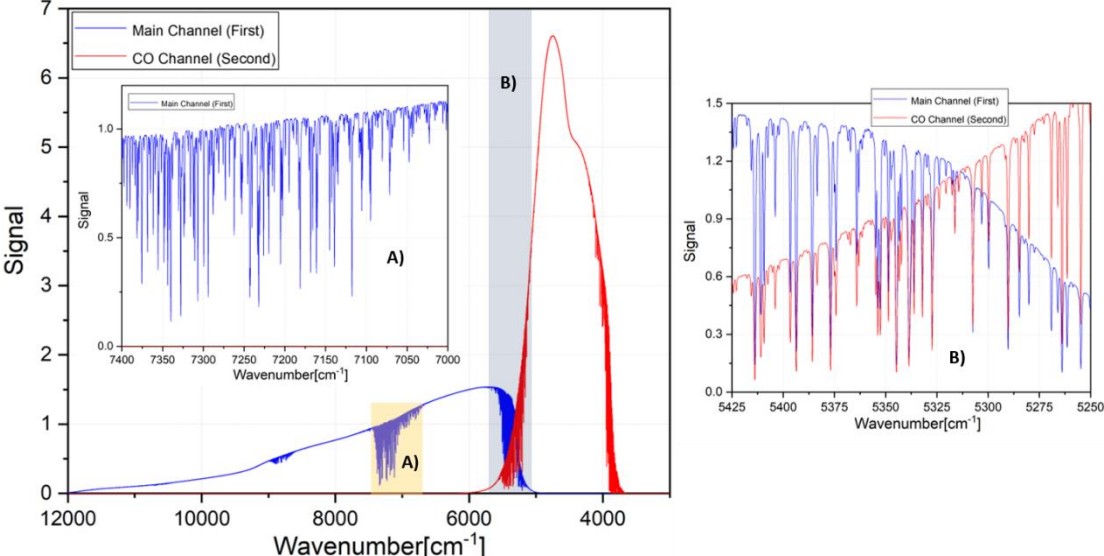

**Figure 3: Typical spectra for both channels, obtained with the COCCON reference instrument SN37 on 19ᵗʰ March 2021. The highlighted regions A) and B) are the spectral windows used for the ILS retrievals.**

### 1.3.3 Empirical update of H₂O spectroscopic data

For the previous ILS analysis, the $H_2O$ line list provided by HITRAN version 2008 (including the corrections introduced in 2009) with some minor empirical adjustments was used. The work presented here uses a considerably revised line list. The HITRAN 2016 $H_2O$ list served as starting point for fitting empirical $H_2O$ line parameters in the two relevant spectral regions using a pair of high-resolution open path spectra recorded with the Bruker IFS125HR spectrometer of the TCCON station Karlsruhe. The air conditioning system of the laboratory container housing the spectrometer was used to adjust the air temperature to 15° C and 30° C, respectively. We assume that this span largely covers the conditions of laboratory ILS measurements. The pair of spectra was then used for a multi-spectrum fit of empirical $H_2O$ line parameters using the LINEFIT software with a wrapper for adjusting the line parameters. Line intensities, line positions, and broadening parameters were adjusted (the ratio of the self and foreign broadening parameters were maintained as reported in the HITRAN line list). The fit residuals of the high-resolution spectra after the empirical adjustment are shown in Appendix A (see Figure 26 and Figure 27); the $H_2O$ line list is provided in the electronic supplement of this work. In order to avoid a significant bias between the ILS parameters reported by Frey et al. (2019) and the results of the reanalysis presented here, a global scaling factor was determined and applied to the new pressure broadening parameters. As expected, the fit quality of EM27/SUN open path spectra using the new empirical line list are significantly improved, as discussed in Section 3.1 .


## 2    Use of a cell filled with a $C_2H_2$-air mixture for ILS characterisation

In addition to the refinements introduced in the open-path method, a new gas cell was developed and has been used in parallel
with the open path measurements. This chapter presents the details of the cell.

### 2.1    Cell components

This new method developed for measuring the ILS for EM27/SUN instruments uses a gas cell filled with $C_2H_2$. This gas is a
good choice, because it is easily accessible and easy to handle, and because it offers a strong absorption band at 6550 cm$^{-1}$, a
spectral region largely free from $H_2O$ contamination. In the context of calibration work for TCCON, experience with $C_2H_2$ has
already been collected (see section 2.2). The cell has an effective length of 200 mm and an internal diameter of 30 mm. Wedged
fused silica windows are glued to the slightly angled end surfaces of the cell body. The cell is closed with a Teflon valve stem
sealing against a Schott Duran valve body. A temperature sensor is attached to the cell in order to monitor this variable during
the experiment. To fix the cell into the lamp beam at the level of the tracker beam, a simple support has been built as shown
in Figure 4 b). A cardboard screen is used to limit the heating of the cell body by the lamp.

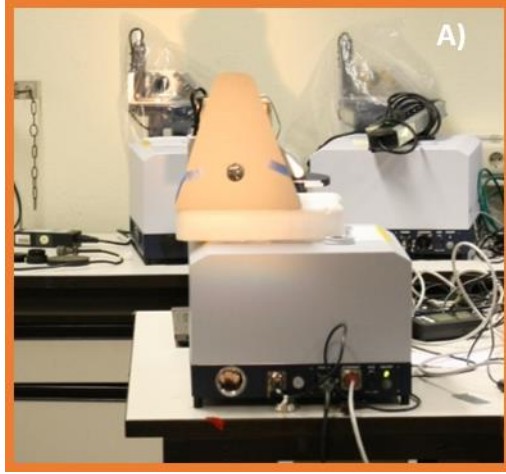
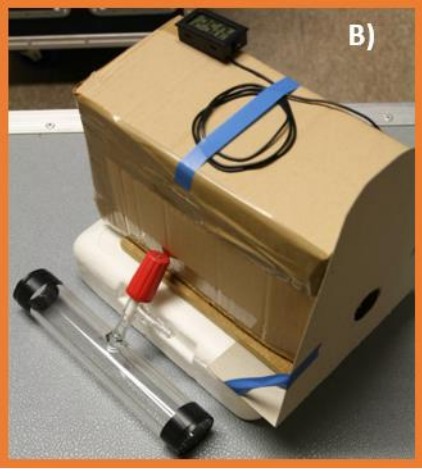


**Figure 4: The set-up of the cell measurements and the cell-components used in this study are shows in A) and B) respectively.**

### 2.2    Cell content and calibration

A different cell, which is pressure-monitored and filled with 300 Pa of pure $C_2H_2$ is used at the TCCON station Karlsruhe for
calibration work on the sealed HCl cell as used by the TCCON network, this cell and its application is described in Hase et al.
(2013). Inspection of the fit residuals of high-resolution $C_2H_2$ spectra recorded with the IFS125HR spectrometer indicate that
especially the line positions of the HITRAN 2016 line list are slightly imperfect, so the line positions have been adjusted. This
improved empirical $C_2H_2$ line list is also applied to the low-resolution work presented here. The empirical $C_2H_2$ line list is
distributed with the LINEFIT code and also provided as a supplement to this paper.



For low-resolution measurements, we require a higher filling pressure, as pressure broadening is needed to generate absorption
lines of sufficient area. In the Doppler limit, even saturated lines generate a very weak signal in the convolved spectrum,
because such lines are spectrally much narrower than the ILS of the EM27/SUN spectrometer. Using an available cell body of
200 mm length, a pressure on the order of 100 hPa was found to be a reasonable choice.

After filling of the cell, a pair of high-resolution reference cell spectra were recorded using the IFS125HR spectrometer at
temperatures around 288 and 303 K. From these spectra, the amount of $C_2H_2$ contained in the cell was retrieved, which also
sets the $C_2H_2$ partial pressure for a given cell temperature. Next, assuming an ideal ILS for the IFS125HR spectrometer, the
relevant cell parameters were retrieved using LINEFIT. The results for $C_2H_2$ total and partial pressure are provided in Table
2. While the partial pressure results from the measured line area follows the ideal gas law, the retrieved total pressure which
minimizes spectral residuals deviates from the ideal gas law. It should not be regarded as a physical parameter, it is used to
compensate for various imperfections (reported values of self and foreign pressure broadening parameters and their
temperature dependence, possible air contamination in the cell, etc.). For adjusting these parameters to other working
temperatures, we apply a linear interpolation in both tabulated parameters of the total and partial pressure.

**Table 2: Measured variables in the cell with respect to the IFS125HR spectrometer at Karlsruhe TCCON station.**

| T [K]  | $p_{tot}$ [hPa] | $p_{part}$ [hPa] |
|--------|-----------------|------------------|
| 288.2  | 138.0           | 121.8            |
| 303.2  | 147.8           | 128.1            |

## 2.3     Measurement setup

When the cell is positioned in the open-path setup, we maintain the four-meter distance between the lamp and spectrometer.
This does not bring in complications, because the $H_2O$ lines superimposed to the observed $C_2H_2$ band are sufficiently weak.
Therefore, we can easily go back and forth between the open-path and cell configuration. The $C_2H_2$ cell introduces a slight
beam deviation because the window wedges do not fully compensate, but the camera incorporated in the EM27/SUN can be
conveniently used for realigning the image of the lamp collimator on the spectrometer's entrance field stop. After the warm-
up phase of the spectrometer discussed in section 1.1, 10 to 16 interferograms are collected using a 10 kHz scan speed (each
interferogram comprised of 10 co-added scans).

## 2.4     Error budget of the cell measurement for measuring ILS parameters of the EM27/SUN spectrometer

With the spectral noise level achieved by applying the measurement procedure outlined in section 3.3, the propagation of
spectral noise into the retrieved ILS parameters is a negligible contribution. We conclude therefore that the error budget is



dominated by the knowledge of the gas cell temperature, which might vary while the measurement is performed and across the cell body. We assume that the knowledge of cell temperature during the measurement is in the order of 1 to 2 K. A change of the temperature by 1 K changes the retrieved modulation efficiency amplitude by about 0.2%.


## 2.5    Data acquisition, pre-and final processing

The OPUS software provided by the manufacturer Bruker is used to collect the interferograms. The settings used for their acquisition are the same as for the open-path method. Once the interferograms are recorded, they are pre-processed using OPUS in the same way as explained in the open-path method, namely a DC correction is included. After generating a

spectrum, the ILS is retrieved using LINEFIT 14.8. Figure 5 shows an open-path spectrum recorded with the $C_2H_2$ cell inserted in the beam. The $C_2H_2$ band located in the wavenumber range 6450 – 6630 cm$^{-1}$ is utilized for the retrieval of ILS parameters.

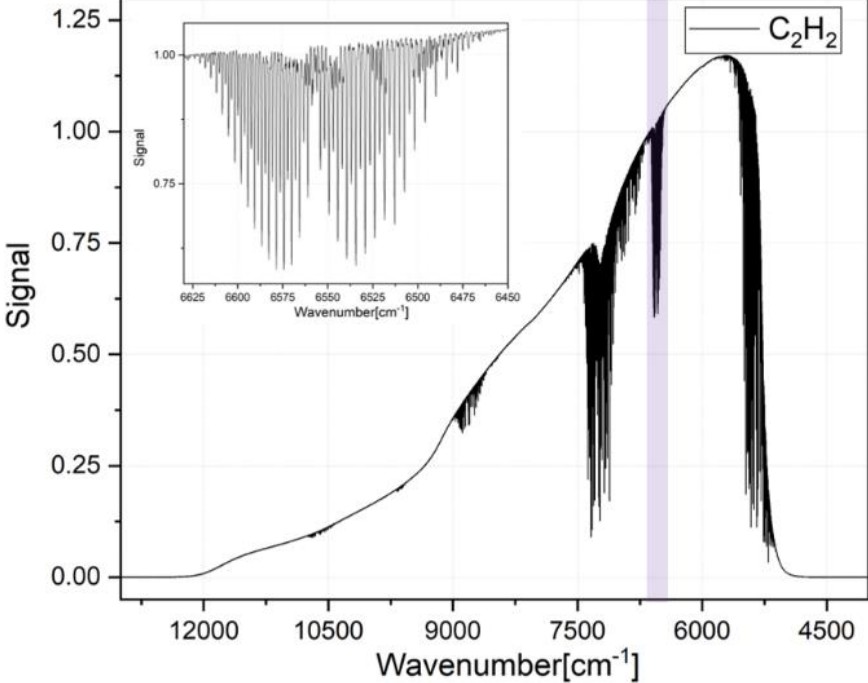

**Figure 5: An open-path spectrum recorded with the $C_2H_2$ cell inserted in the beam. The spectrum was recorded using COCCON's**
**EM27/SUN reference spectrometer SN37. The insert shows a zoom-in of the wavenumber range used for the retrieval of ILS parameters from $C_2H_2$.**





## 3 Discussion of open path results

In this section, we will discuss the results of the open path measurements, achieved by applying the improved and extended methods introduced in Section 1. Firstly, we apply the new refined analysis procedure to the open path measurements collected

by Frey et al. (2019), and we compare the results of this reanalysis with the previously reported results. Next, the results derived from the standard spectral window are compared with results obtained using the micro window in the spectral overlap region, which is accessible by both detectors. As described in Section 1.3, this additional spectral micro window was implemented for detecting a potential misalignment of the CO detector element. This performance test was not included in the previous open-path recipe. Finally, our best estimate of the instrumental line shape parameters is provided for all tested spectrometers. The

table summarizing the revised results contains the revised values for those spectrometers investigated in the study by Frey et al. (2019), and new results for the spectrometers, which have been calibrated since then.

### 3.1 Reanalysis of previous open path measurements

Figure 6 shows for all spectrometers treated in the work of Frey et al. (2019), the old and newly derived modulation efficiency

amplitudes (MEA), the phase errors (PE), the new-minus-old differences for both quantities, and the empirical standard deviation of the spectral residuals. The use of the revised $H_2O$ line list significantly reduced the spectral residuals. Figure 7 shows an excellent correlation between old and new MEA ($R^2 = 0.94$) and PE ($R^2 = 0.93$) results. Figure 8, left panel, shows that due to the empirical calibration of the $H_2O$ broadening parameters mentioned in section 1, only a small bias in MEA is seen, the mean of the new MEA results being lower by 0.24%. Figure 8, right panel, indicates a significant reduction of PE

values, so probably part of the previously diagnosed ILS asymmetry was introduced by systematic spectral residuals created by the HITRAN 2008 line parameters. Overall, the revised analysis recipe confirms the results by Frey et al. (2019), as spectrometers showing suspiciously high or low values of MEA or PE versus the average behaviour retain their characteristics. Although we are confident that the new method, using an improved line list, a correction of the optical distance (and thereby $H_2O$ self-broadening effects), and more reliable data for the total pressure, is superior to the original method, the overall effect

is only a gradual improvement. The reanalysis of the old spectra is important mainly in order to avoid a systematic bias of reported ILS parameters between previous and current calibrations.



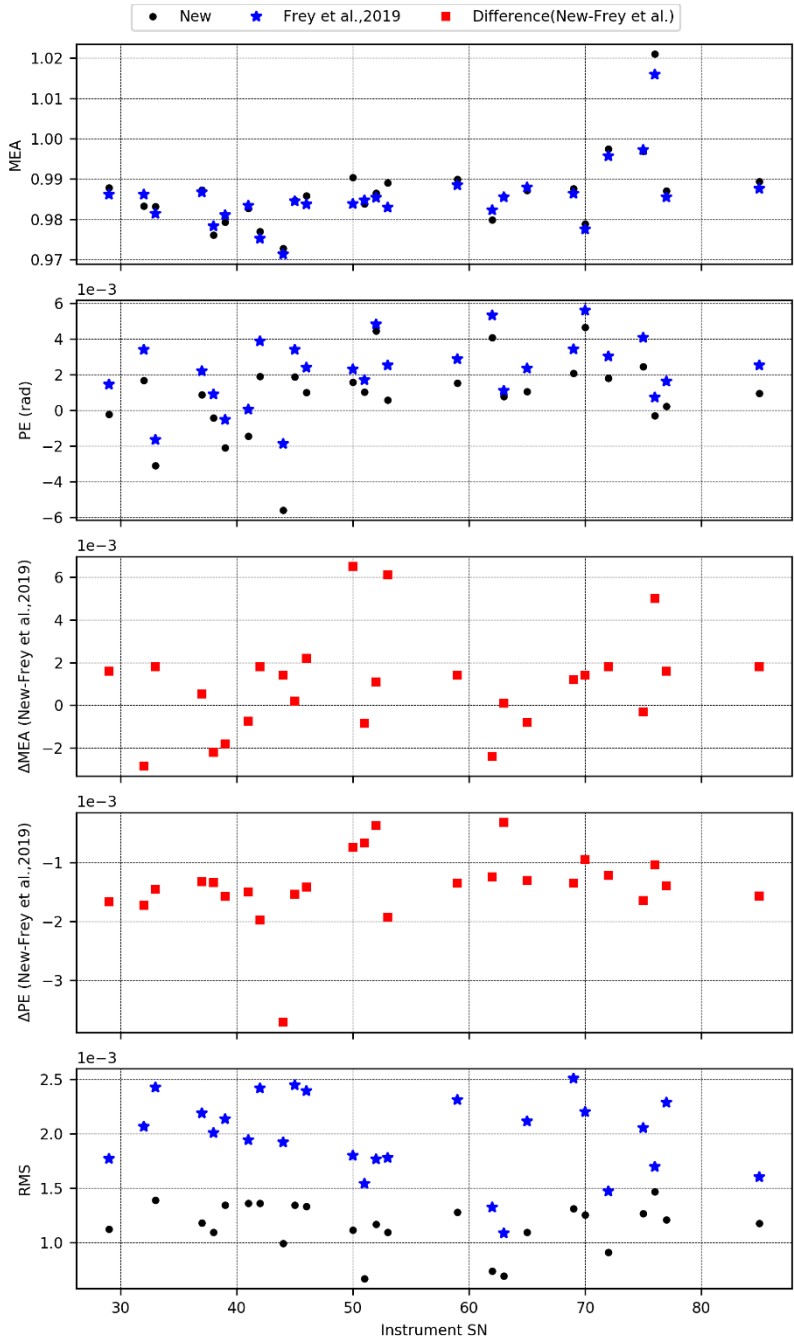

**Figure** 6**: Comparison between the old published values (blue star) and the improved ones (black dotes). The MEA, PE and the new-minus-old difference for each spectrometer. The bottom panel shows the resulting empirical standard deviation of the spectral fit for the old and the new methods, respectively.**






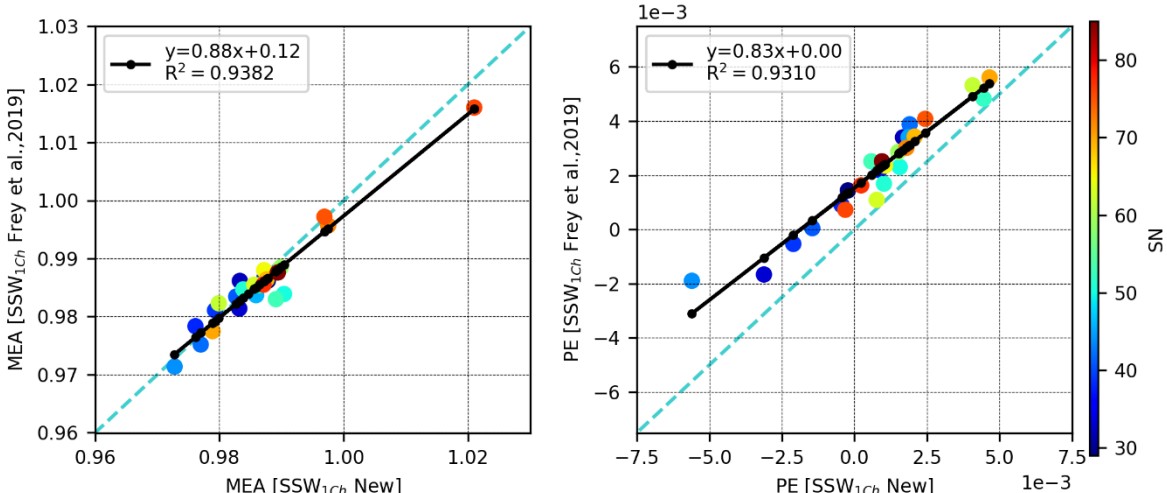

**Figure 7: Left panel: correlation between the MEA obtained with the new and the old methods for the shortwave standard micro window (SSW). Right panel: correlation between the PE obtained with the new and the old method. The colour bar represents the SN of the instruments.**

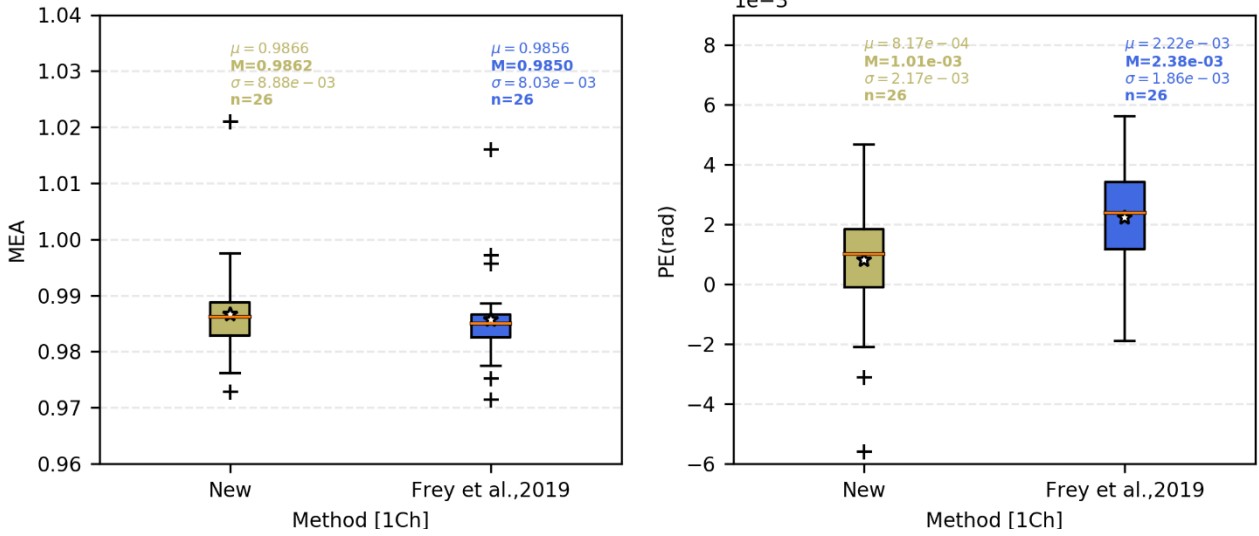

**Figure 8: Box-and-whisker plots showing the statistics of the original data analysis by Frey et al., 2019 and the reanalysis: median, mean, scatter, and interquartile range are presented. Left panel: MEA, right panel: PE**

### 3.2    Open path results for all spectrometers

In this Section, the ILS parameters for all spectrometers as retrieved with the improved analysis procedure are presented. The left panel of the Figure 9 provides a graphical overview of these new results, including the reanalysis results for the spectrometers already investigated by Frey et al. (2019). In total 47 new spectrometers were investigated. As can be seen from the Figure, the results for new spectrometers are in line with the previous work, but the occurrence of outliers seems reduced



(the clearly deviating behaviour of spectrometers 75 and 76 uncovered by the calibration work was later diagnosed to be caused by misassembled detector baseplates). Presumably this reflects the gain of expert knowledge in the fabrication of the EM27/SUN spectrometer type and in the acceptance and calibration procedures. We suppose that the continued efforts for quality assurance presented in this work contribute to the high level of consistency achieved in the spectrometers' characteristics that is apparent today. Table S1 in the supplement of this paper collects the ILS results for all spectrometers.

## 3.3 Testing the alignment of the CO channel


The addition of a further spectral window to the open-path analysis in the spectral overlap region covered by both the main and the CO channel allows the extension of the open path ILS analysis to the CO channel. The CO channel is an extension of the original design of the EM27/SUN (Hase et al., 2016). CO is an air pollutant, and also useful for the source apportionment of $CO_2$ emissions. CO is measured by space sensors as Measurement of Pollution in the Troposphere (MOPITT) (Drummond, 335 1992; Drummond and Mand, 1996) and the TROPOspheric Monitoring Instrument (TROPOMI) (Veefkind et al., 2012). Today, all EM27/SUN spectrometers incorporate both detector channels. Therefore, it is desirable to include a procedure in the calibration which recognizes the potential for a significant misalignment of the CO detector element with respect to the main detector. Such a misalignment of the CO detector would generate (1) deviating ILS parameters and (2) a deviating spectral scaling factor.


In this Section we compare the consistency of spectral fits in the standard spectral window (SSW) and in the overlap region (OVR) using the spectra recorded with the main detector. We compare the retrieved ILS parameters (MEA and PE) and spectral scaling factors. In section 3.3.1, we investigate the consistency of ILS results between the SSW and OVR windows using spectra recorded with the main detector channel.

In Section 3.3.2, we discuss the results from the OVR region, this time using spectra recorded with the CO detector instead of the main detector.





**Figure 9: Main results for the main (left panels) and CO detector channels (right panels) resulting from the revised open-path**
**method. The modulation efficiency, phase error, RMS and relative difference for the first channel by using SSW and OVR is shown**
**in a), c), e) and g) respectively, while the modulation efficiency, phase error, RMS and relative difference for the first and second**
**channel using SSW and OVR is presented in b), d), f) and h) respectively.**





**Figure 10: Correlation plots between the MEAs and PEs (in the left and right panel respectively) : panel A: the first channel in the SSW and in the OVR region, panel B: the first channel in the SSW and the second channel in the OVR region, and panel C the first channel in the OVR and the second channel in the OV R region . Additionally, the obvious outliers are labelled in order to assess them.**





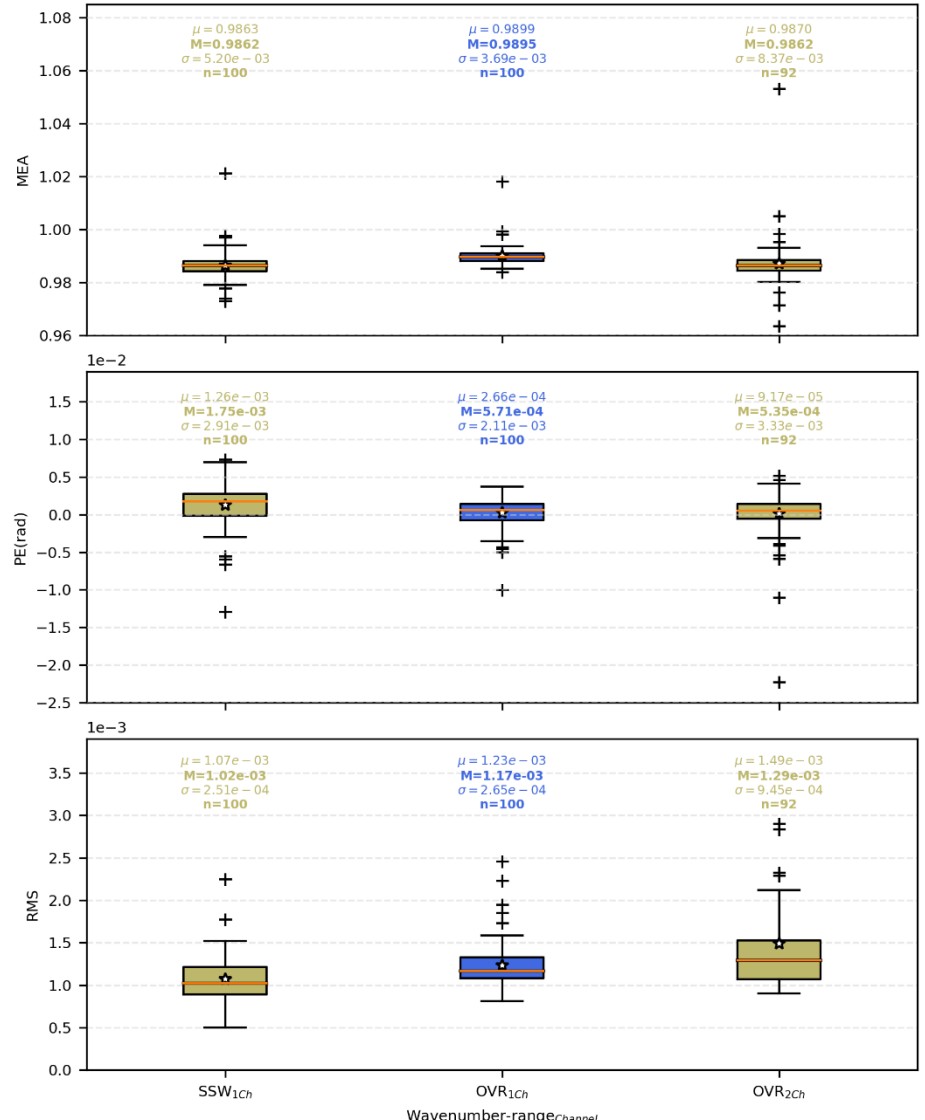

**Figure 11: Box plots comparison for the three wavenumber ranges used with the open-path method showing the MEAs, PEs and**
**RMS**

### 3.3.1 Consistency of spectral fitting in the standard spectral window and the overlap region

Figure 10 (top panel) compares the MEA and PE of retrievals performed in the SSW and in the OVR using the main detector. The results show good agreement (MEA: $R^2 = 0.78$, PE = 0.93). It is very interesting to observe that the regression line has a slope significantly below 1:1. Since parameters such as MEA and PE measure fractional wave front errors, their deviations from the nominal value are indeed expected to increase with increasing wavenumber. The wavenumber ratio between OVR and SSW is 0.74, while the slope of the MEA regression line is 0.63, which would support the assumption of a steeper than





linear wavenumber dependence of the MEA parameter ( $\sim \nu^{1.5}$ ). The PE results are compatible with the assumption of a linear wavenumber dependence.

As can be seen from Figure 11, there is a small bias of 0.3% in MEA: the values retrieved in the OVR are slightly higher than
those from the SSW are. The PE retrieved in the OVR is significantly smaller, this might indicate that the revised spectroscopic description of the SSW spectral window – although the new line list reduced the retrieved PE by a factor of two (see section 3.1) – still simulates a spurious PE bias-.

Figure 12 summarizes the results for the spectral scaling factors for both spectral windows as resulting from the LINEFIT fits. Figure 13, left panel, compares the spectral scaling factors of OVR and SSW fits as deduced from main detector spectra. As
one would expect, the slope is near to 1:1 and the correlation is very high.

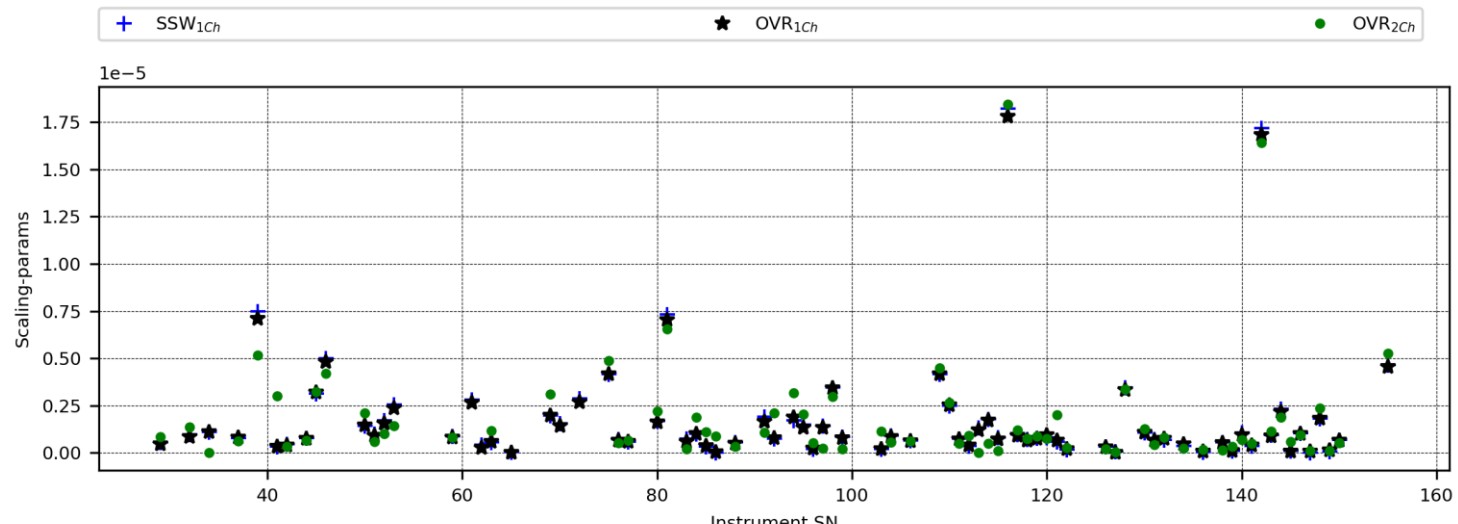

**Figure 12: Instrumental variation of the spectral scaling factors in each of the spectral windows used and for both channels.**




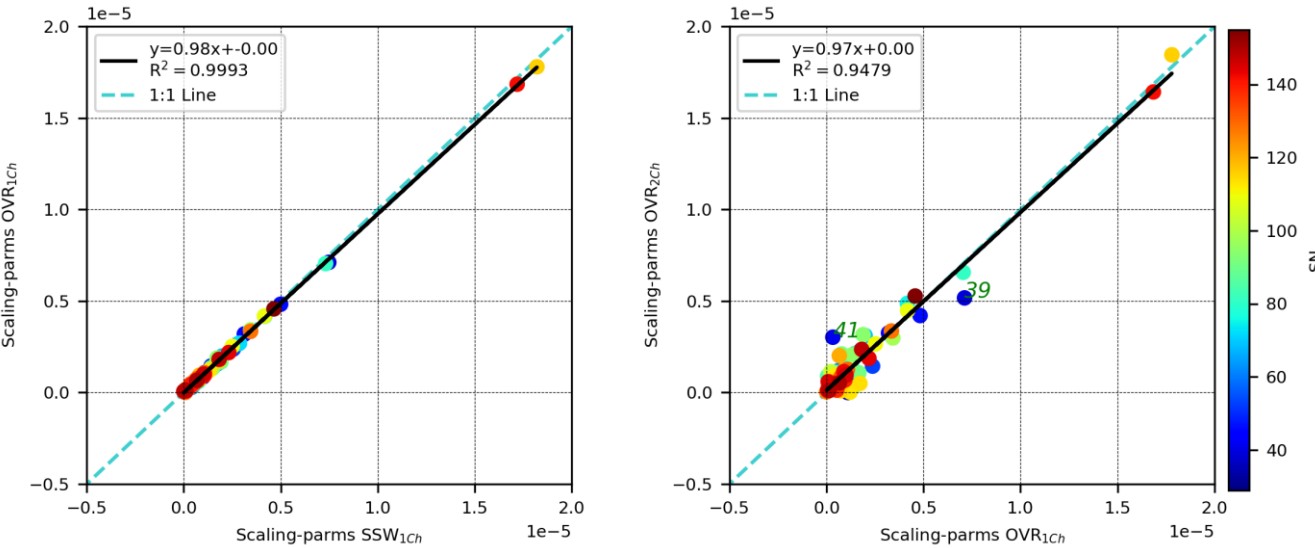


**Figure 13: Correlations between the scaling factors derived from OVR and SSW spectral windows using the main channel spectra (left panel), and using CO channel results for the OVR spectral window (right panel).**

### 3.3.2 Evaluation of the CO detector alignment using the spectral overlap region

Figure 10 (middle and lower panel) shows the MEA and PE correlations as deduced from the main detector and the CO detector, respectively. Figure 10, middle panel, shows the correlation of MEA between the CO detector (OVR) and main detector (SSW). While the shallower slope is comparable with the results reported in Section 3.3.1, the correlation between the two different detectors is significantly poorer. There are several outliers from the MEA regression line: these are spectrometers SN: 39, 42, 53, 75, and 110. In the PE regression, the five results furthest from the regression line are SN: 50,

94, 110, 111 and 143.

Figure 13, right panel, compares the spectral scaling factors of OVR fits for the two CO and main detector. While the slope is in excellent agreement with the results derived from the main detector, there is more scatter ($R^2 = 0.95$). The results for spectrometers 39 and 41 are furthest from the regression line.


In summary, although the correlation of ILS parameters and spectral scaling factors is noisier between the main and CO detector, only one consistent outlier appears, which is spectrometer SN39. Altogether the applied OVR tests do not detect unacceptable misalignments of the CO detector. The relative spectral detuning of SN39 between SSW and OVR is in the order of $2 \times 10^{-6}$, which, by applying $\frac{\Delta v}{v} = \frac{1}{2} \alpha^2 \, using \, \alpha \sim 1.5 \, mrad$, is equivalent to about 1/7 of the apparent solar diameter. Here $\alpha$





denotes the maximum inclination of a ray still accepted by the interferometer. The effect of a misadjusted field stop on spectral scale is discussed by Kauppinen and Saarinen, 1992. The majority of spectral detuning results is located within $\pm 1.5 \times 10^{-6}$, equivalent to an angular misalignment of 1/14 of the apparent solar diameter, which is in reasonable agreement with the expected alignment precision of the CO detector. Because the airmass reference is deduced from the oxygen band observed in the main channel, such a misalignment introduces an error in the XCO data. If we assume a misalignment of 1/10 of the

apparent solar disc diameter along the vertical, the resulting relative error in XCO at 80° solar zenith angle amounts to 0.5 %.

## 4      Discussion of $C_2H_2$ cell results

In Section 2, the construction and calibration of a cell filled with $C_2H_2$ is described. Here, we compare in detail the results obtained from the open-path measurements (OP) using the $H_2O$ lines forming in the laboratory air with those obtained with the cell method. Because the cell measurements were implemented in the beginning of 2020, only spectrometers tested

afterwards are cell results available. The comparison is based on the standard $H_2O$ window covering 7000-7400 cm$^{-1}$ discussed in section 1 and the $C_2H_2$ spectral window covering 6450 - 6630 cm$^{-1}$ discussed in section 2, so spectra recorded with the main detector are used.

### 4.1      Intercomparison of repeated open path and cell measurements using the reference spectrometer

In order to investigate the stability of both methods, OP and cell measurements were taken repeatedly under different laboratory

conditions using the COCCON reference instrument SN37. On a total of 16 days, measurements were performed during February and March 2021. For each daily set of measurements included sequential OP and cell measurements were taken within 45 minutes to ensure the laboratory conditions were comparable. We collected 15 interferograms for the cell test and 30 for the OP method.

Figure 14 shows the internal variability of the results. Both methods seem to offer similar repeatability. While we do not see a clear advantage of the cell approach from the comparison in this regard, we need to acknowledge the fact that the $C_2H_2$ line widths are properly calibrated. If we assume that the TCCON spectrometer used to calibrate the empirical $C_2H_2$ cell parameters can be regarded as an ideal reference (see Section 2.2), this finding suggests that the OP MEA results indeed suffer from a systematic low bias of about 0.015 (1.5%) and that the ILS performance of the EM27/SUN is on average closer to the nominal

expectation than indicated by the OP measurements (see Figure 15). This adjustment will be included in a future version of the PROFFAST software used by COCCON for the analysis of atmospheric spectra. The current version of the code uses the MEA values resulting from the OP measurements, so the currently incorporated values of the airmass-independent and airmass-dependent calibrations are partly mitigating the bias in MEA.





Even though the cell method does not provide a significant improvement in the determination of MEA and PE values, we plan

to maintain the cell measurements in the calibration procedure. That the cell measurement delivers a column value, which can

be measured with excellent precision and provides an invariant for the comparison of different spectrometers, seems a useful

addition. The relative one-sigma standard deviation of the $C_2H_2$ column indicated by the repeated measurements is 0.01%

(individual column results are shown in Figure 14 C)).

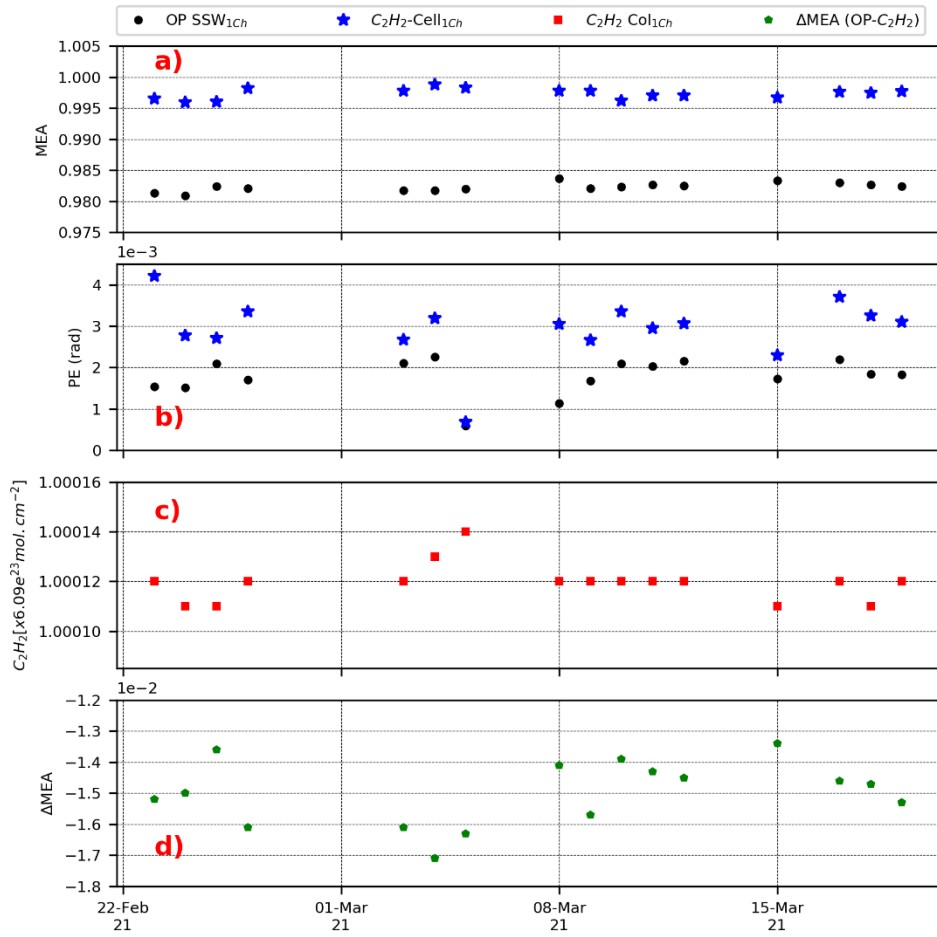


**Figure 14: Time series of the MEAs, PEs, $C_2H_2$ retrieved column and difference between the MEAs retrieved with OP and cell method for the COCCON reference instrument SN37 a), b), c) and d) respectively.**



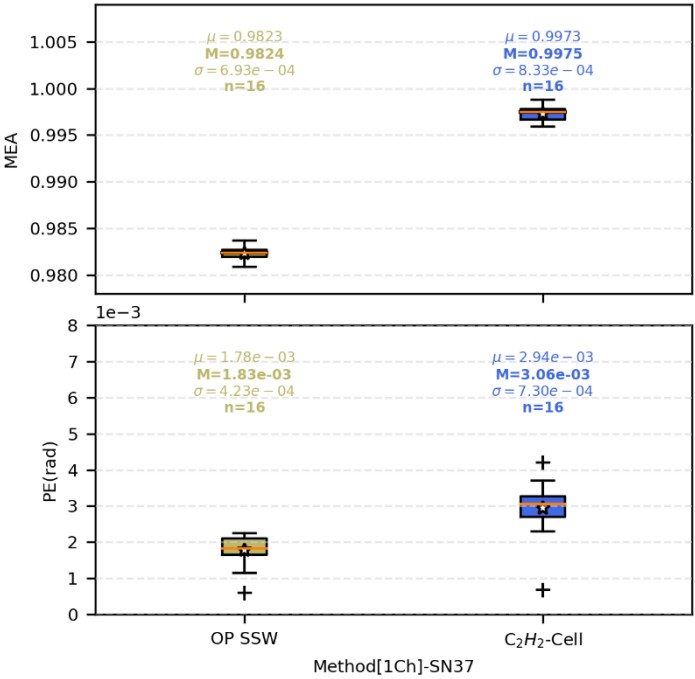

**Figure 15: Same as Figure 8 but for the sensitivity study for the COCCON reference instrument SN37. Left part of the display: open
path results, right part: cell results.**

## 4.2    Intercomparison of cell results with open path results

This Section summarizes the main results of the ILS characterization for the first channel by using the OP and the cell method

for the spectrometers tested since 2020 (see Figure 16, Figure 17 and Figure 18). Figure 16 show the instrumental variation of

the MEA and PE, RMS according to both methods.   The MEA retrieved with the cell method is higher and closer to the ideal

ILS in comparison with the OP method, which supports the finding discussed in the previous section that the cell method

retrieves ~ 1.5% higher MEA values. Figure 17 shows the correlation between the OP and the cell MEA and PE results and

figure 18 shows a statistical comparison. We find a reasonable correlation, which indicates that despite the tendency that the

spectrometers become more uniform in their characteristics, we still are able to detect - using the described laboratory

procedures - actual variations of the MEA and PE values. The sensitivities differ between the methods: while the slope of the

MEA regression line is compatible with our assumption of a $\sim \nu^{1.5}$ wavenumber dependence of the MEA parameter (see

discussion in Section 3.3.2), the slope of the PE regression line is surprisingly steep, as we would expect PE to be proportional

to wavenumber. However, the spectral scenes are quite different; the $C_2H_2$ lines offer a significantly smaller width than the

$H_2O$ lines. Therefore, the ILS deviations associated with contributions emerging from larger optical path difference (OPD)

will gain importance in the $C_2H_2$ spectral fitting. The assumption of a constant PE might be too coarse and therefore introduces





the observed discrepancy between the two methods. When regarded from this perspective, the continuation of the $C_2H_2$ measurements in addition to OP might also become useful for introducing further refinements of the ILS model in the future. Figure 18 summarizes the performance of both methods.

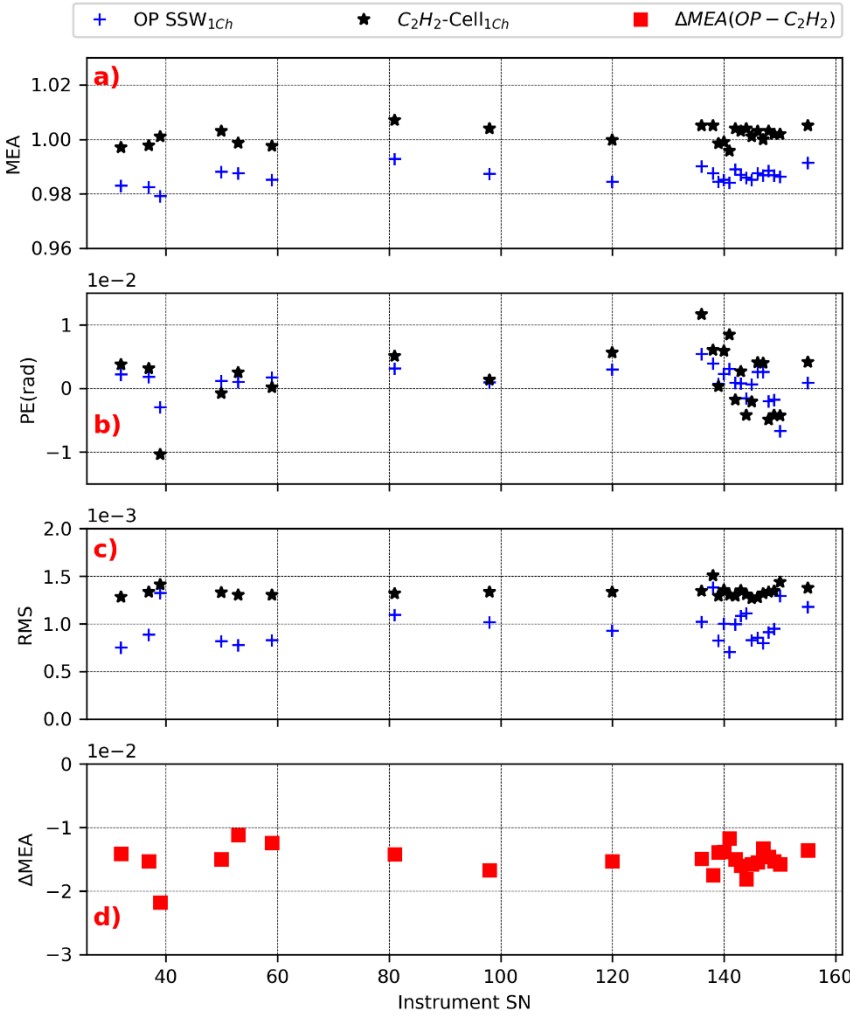

**Figure 16: The modulation efficiency as function of the instrumental SN, phase error, RMS of the spectral fit and the relative difference between the open path and the cell method are presented in a), b), c) and d) respectively.**





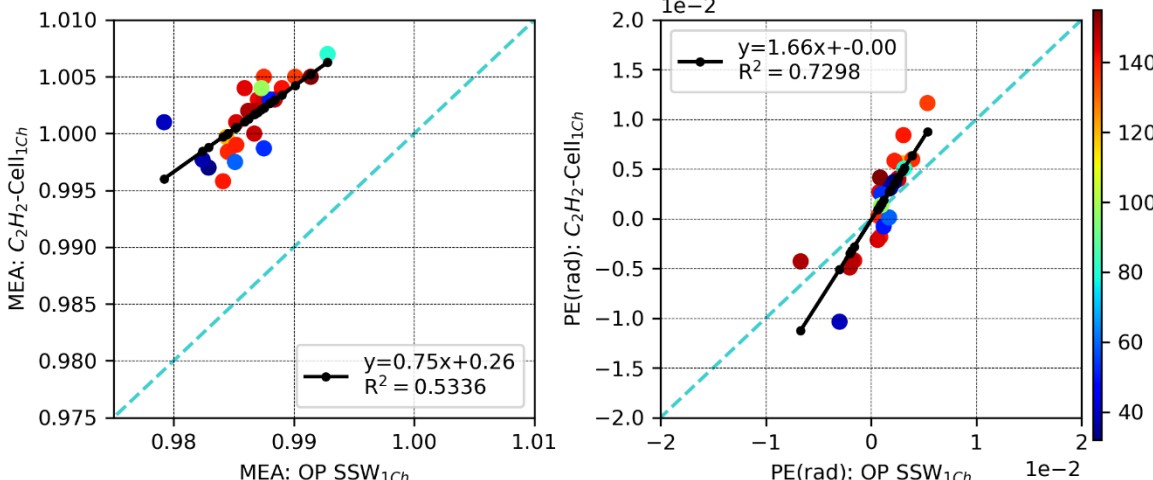

**Figure 17: Correlations between the MEAs obtained with the OP and cell method for the first channel**

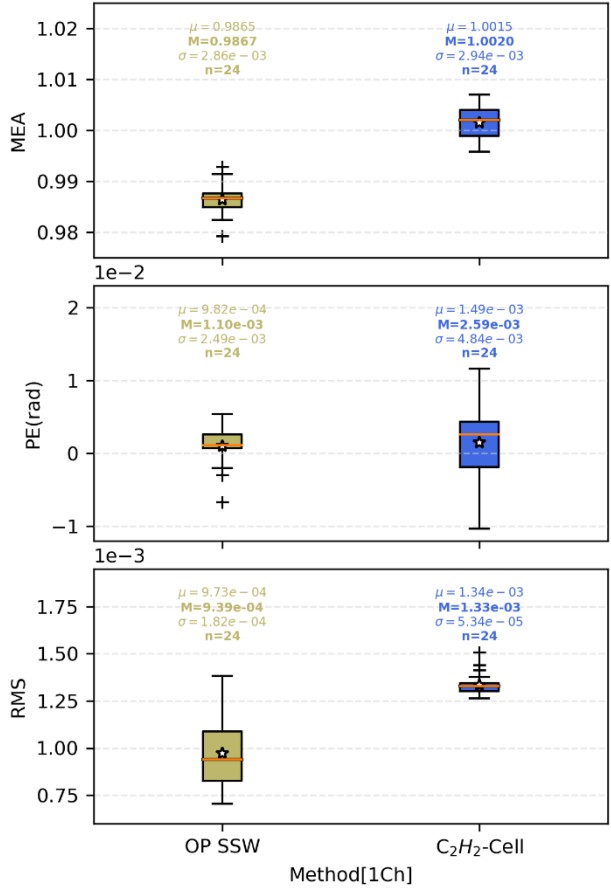

**Figure 18: MEA and RMS statistical resume from the ILS retrievals by using the OP and the cell method for the first channel of the available instruments (left part of the display: open path; right part: cell).**





## 5  Discussion of solar side-by-side calibration measurements

### 5.1  Long-term stability of reference unit

In this section the historic time series of the COCCON reference instrument SN37 is assessed by comparing the main target
gases: XCO₂, XCO, XCH₄, and XH₂O with the results obtained from the high-resolution spectrometer IFS125HR located at
KIT Campus North (49°06'00.8"N, 8°26'18.6"E, 112 masl). This spectrometer contributes to the TCCON network. Two
different kinds of measurements were collected with the IFS125HR spectrometer: standard TCCON measurements using a
spectral resolution equivalent to max. OPD of 45 cm and double-sided low-resolution spectra for mimicking the EM27/SUN
observations (maximum OPD 1.8 cm). The COCCON and the low resolution data recorded with the IFS125HR were analysed
using PROFFAST, while the high-resolution spectra were used for generating the official TCCON product using the GGG
software suite version 2014 (Wunch et al., 2015). Because it provides a very sensitive indication for instrumental drifts and
operation problems, we also investigate here results for X_AIR. This quantity compares the spectroscopically determined dry air
column as extrapolated from the observed vertical column of O₂ $VC_{O_2}$ with the dry air column calculated from ground pressure
and spectroscopically observed water vapour column $VC_{H_2O}$, as given in Eq. 1.

$$X_{AIR} = \frac{0.2095}{VC_{O_2}.\bar{\mu}}.\left(\frac{P_S}{g} - VC_{H_2O}.\mu H_2O\right)$$   Eq. 1

### 5.1.1  PROFFAST code and COCCON reference EM27/SUN spectrometer

The initial development and further improvements of the PROFFAST code are supported by the European Space Agency
(ESA) in the framework of the COCCON-PROCEEDS project. The code aims at efficient analysis of greenhouse gases from
ground-based near infrared solar absorption spectra. Together with the pre-processing code PREPROCESS, it forms the data
analysis chain of COCCON. The code is open-source and freely available. It performs least squares fitting of the spectra by
adjusting scaling factors on the a-priori profiles of the trace gases and auxiliary parameters. It is important in the context of
this work that PROFFAST is capable of taking into account the ILS parameters as determined by the open-path measurements.
If this information is neglected, additional scatter between the atmospheric trace gas results achieved with different
spectrometers would result and different gas-specific empirical calibration factors would result from the side-by-side solar
observations for each spectrometer (these factors are reported in section 5.2). Additional information on the code is provided
by Frey et al. (2021) and Sha et al. (2020).

The EM27/SUN spectrometer SN37 has served as the COCCON reference spectrometer since 2014. The spectrometer
participated in the Berlin campaign (Hase et at., 2015) and was upgraded with the CO channel in early 2018. Figure 19 presents
the time series of XCO₂, XCO, XCH₄, XH₂O and X_AIR covering 2015 to end of 2020. Shown are the official TCCON data



generated with the GGG2014 software suite and data derived from the low-resolution spectra recorded with the IFS125HR spectrometer and the COCCON reference spectrometer, respectively, using the PROFFAST code. For the target gases, no obvious drifts are noticeable between the different data sets. The bias in $X_{AIR}$ between the TCCON and low-resolution data is

due to the trivial fact that $X_{AIR}$ is not generated as a calibrated quantity by GGG2014, while PROFFAST attempts a normalization to unity. However, there is a change of $X_{AIR}$ apparent in the COCCON reference data during the first four years, which we investigate further in the next section. We will show that these changes are small enough not to detectably affect the results of the target gases apart from $XCH_4$.



**Figure 19: Time series of XCO₂, XCO, XCH4, XH₂O and X_AIR measured with the COCCON reference instrument (blue), from the TCCON station Karlsruhe (derived from high-resolution IFS125-LR spectra using GGG2014, red) and derived from low-resolution IFS125-LR spectra (black). The low-resolution measurements were processed with PROFFAST.**





**Figure 20: Correlations between XCO₂, XCO, XCH₄ and XH₂O between the ones retrieved by using the COCCON reference and the IFS125-LR low-resolution data (left panels), and between COCCON reference and TCCON station (right panels).**





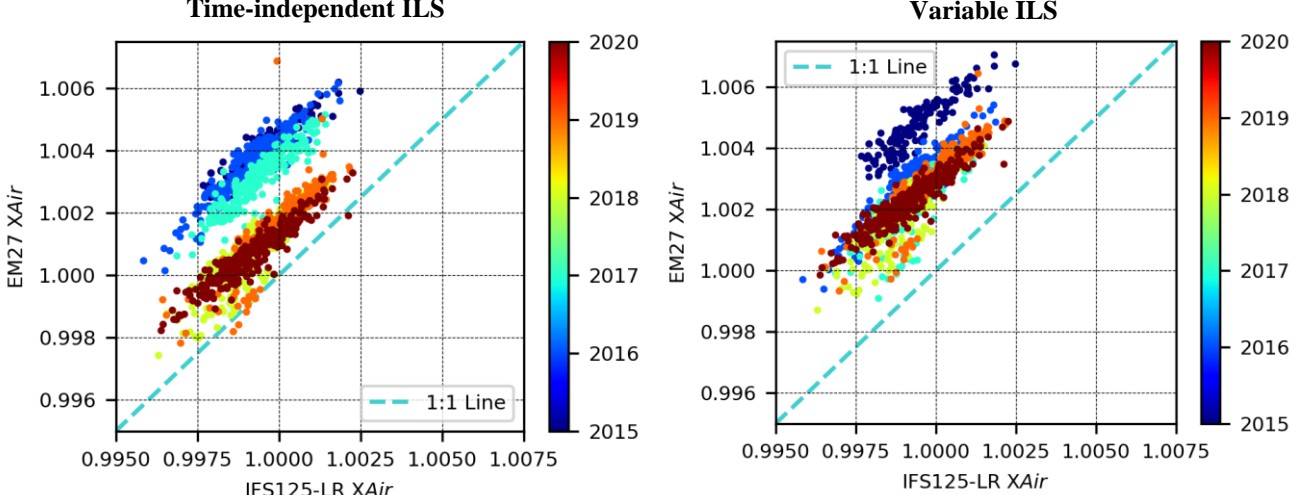

**Figure 21: Correlations for the retrieved X_AIR by using the instrument SN37 and IFS125-LR. The left panel shows the results of the analysis of atmospheric spectra under the assumption of a constant ILS, the right panel shows the results under the assumption of a variable ILS (ILS parameters adjusted on a yearly basis).**




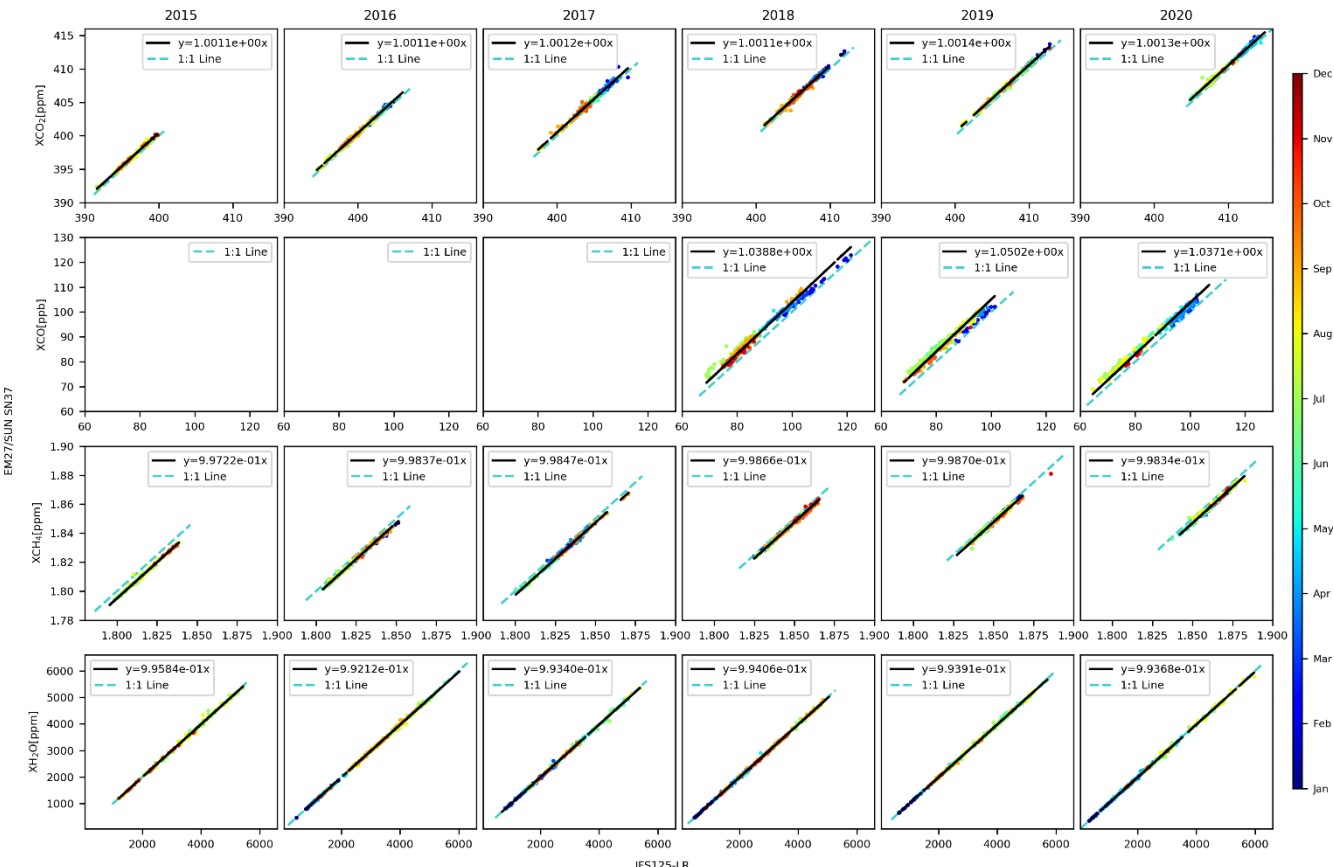

**Figure 22: Correlations between the species: XCO₂, XCO, XCH₄ and XH₂O-retrieved with the COCCON reference instrument and the TCCON instrument in low resolution measurement mode in each row-top-down respectively treated separately by year from 2015 to 2020 in each column for each species.**


### 5.1.2 Changes of X$_{AIR}$ in time series of reference spectrometer

Figure 21 shows the variations of X$_{AIR}$ of the COCCON reference unit with respect to the low-resolution IFS125HR data. At least two step-changes appear, at the end of 2015 and at the end of 2017. Since 2018, the results appear stable. The step-change end of 2017 is very likely associated with the CO channel upgrade of the spectrometer, while the earlier event might be

associated with a realignment of the spectrometer performed in the winter period after participation of the unit in the Berlin campaign between June and July 2014 (Hase et al., 2015). The analysis of atmospheric spectra collected with the reference unit was performed twice: in one analysis, it was assumed that the ILS is time independent (the ILS parameters used for the analysis were derived from averaging the parameters from all available ILS measurements performed with the reference spectrometer). In the other analysis, yearly values for the ILS parameters were applied as deduced from the available open-





path measurements. With the exception of 2015, the $X_{AIR}$ results appear more consistent if time-dependent ILS parameters are used for the data analysis. In 2015, only a single ILS measurement was performed, and might for some reason be of inferior quality. A reanalysis of the open-path spectra uncovered at least the use of an erroneous ground pressure value in the original analysis of this measurement reported by Frey et al. (2019), and resulted in less anomalous values for the ILS parameters. The revised set of values (MEA = 0.98417 and 0.98430 and PE = -0.00061 and -0.00068 instead of MEA = 0.98555 and 0.98940

and PE = -0.00086 and 0.08658) has been used in the current analysis for the 2014 and 2015 period, but the MEA value is still suspiciously high. The OP procedures were less refined in the beginning (e.g. no venting of the spectrometer was performed), so the measurements are less reliable than current OP measurements.

The consideration of the variable ILS brings the $X_{AIR}$ results from 2016 and 2017 in better agreement with the more recent results, with only the 2014 to 2015 period remaining an outlier. We therefore conclude that the assumption, that real ILS

changes occurred in the early years due to instrumental interventions and upgrades is the best choice. The results shown in Figure 19, Figure 20, and Figure 22 all refer to the analysis run using the variable ILS parameters.

Figure 20 and Figure 22 investigate the correlation of the retrieved dry-air mole fractions between the reference spectrometer and the data derived from IFS125-LR measurements. While no significant changes are detectable for $XCO_2$, XCO, and $XH_2O$, the $XCH_4$ regression line in Figure 20 is shallower than the 1:1 line. Figure 22 investigates the correlation year-by-year. Again,

the changes for $XCH_4$ become apparent. We therefore assume for the $XCH_4$ time series from the COCCON reference unit the existence of a non-negligible drift over the first years. We assume that the reference spectrometer has reached a stable configuration since 2018 and during this period we use the $XCH_4$ side-by-side results without further corrections. Before this period, we derive from Figure 22 the existence of a low bias of the reference unit and therefore apply during 2017 a low $XCH_4$ bias of the reference unit of 0.0001, in 2016 of 0.0002, and in 2015 of 0.00135 (relative detuning of $XCH_4$ calibration). The

instrument specific $XCH_4$ calibration factors provided in section 5.2 and in Table S2 in the supplement of this paper take these corrections of the reference unit into account.

The variable bias of the reference unit's $XCH_4$ despite the fact that a time-dependent ILS is used in the data analysis might indicate that the ILS model currently used by PROFFAST is too simple or that the assumptions made on the wavenumber dependence of the ILS parameters are incorrect (the current version of PROFFAST assumes a linear wavenumber dependence

for MEA and PE while the open-path analysis suggests a quadratic dependence for MEA, see Section 3.3.1), or that additional influencing factors are affecting the trace gas results.

## 5.2 Empirical XGAS calibration factors for all tested spectrometers

To harmonise the retrieved species when using any COCCON spectrometer, empirical instrument-specific calibration factors

for $XCO_2$, XCO, $XCH_4$ and $XH_2O$ are calculated from the side-by-side solar measurements with the reference spectrometer SN37. The instruments are set-up on the seventh floor at the Meteorology and Climate Research - Atmospheric Trace Gases and Remote Sensing (IMK-ASF) building located at KIT campus north (49°05'38.7"N 8°26'11.5"E, 134 masl). After the





measurements are taken, the data are processed using the PROFFAST software. In this processing, the ILS parameters derived previously from OP measurements are included for both spectrometers, the spectrometer under test and the reference unit.

Ideally, the resulting gas abundances measured by the spectrometers would be found to be identical. The residual biases give rise to instrument-specific empirical calibration factors that are reported in the following for each spectrometer and target gas. These empirical adjustments consider all remaining instrumental imperfections which are not properly quantified in the calibration process or not properly reflected in the trace gas analysis.

The correction factors are defined in Eq. 2 where the $K_{gas}^{SN}$ is the correction factor and $X_{gas}^{no-corr}$ is the dry air amount of a defined gas without any correction for a defined gas and instrument. The correction factors are calculated by comparing a defined gas retrieved with any EM27/SUN instrument with the reference instrument; a linear fit forced to zero intercept is performed and then the slope is taken as its value.

$$X_{gas}^{corr} = K_{gas}^{SN} * X_{gas}^{no-corr}$$ Eq. 2


Figure 23 shows and lists the empirical calibration factors for $XCO_2$, $XCH_4$, XCO, and $XH_2O$ for each spectrometer investigated. Several spectrometers were calibrated repeatedly, in such cases the values are mean values (the individual results are provided in Table S2 in the supplement of this paper). The table also provides the $X_{AIR}$ value for each spectrometer. While the Xgas values are derived from the measurements taken with the spectrometer under test and the reference unit, the $X_{AIR}$

result is independent from the reference unit.

Figure 23 provides a graphical overview of the tabulated values. One-sigma error bars are shown, if several calibrations were performed on a spectrometer. Similar to what has been observed and discussed before for the ILS parameters (see Section 4.1); a trend towards improved consistency of the calibration factors is suggested, especially for $XCO_2$ and $XCH_4$. XCO is a very weak absorber and therefore the scatter is largely dominated by residual channelling (Blumenstock et al., 2021), which

continues to show variable characteristics between individual spectrometers.



(Figure 23 scatter/error-bar plot)

**Figure 23: Correction factors for XCO₂, XCH₄, XCO and XH₂O from left to right respectively, calculated for all EM27/SUN spectrometers, the error bar represents the standard deviation and it is shown only for the instruments with more than one side-by-side measurements in Karlsruhe. The dashed line represents the ideal value '1.0' (practically realized by the COCCON reference spectrometer SN37).**






### 5.3 Spectral signal-to-noise ratio of the EM27/SUN spectrometers

In order to assess the distribution of the spectral signal-to-noise ratio SNR of different instruments, these values were calculated for both solar and laboratory spectra. For both cases the SNR is calculated for several spectral windows covering both detector channels. The procedure applied is based on the formula described in Bruker OPUS © software manual (2018); the SNR is calculated from the ratio of two consecutively measured spectra. A wavenumber region largely free of absorption gases lines is selected. A parabola is fitted to the ratio spectrum in the investigated spectral window and serves as nominal signal. The

RMS of the fit residuals is calculated. This RMS value is divided by $\sqrt{2}$ to deliver the SNR of a single spectrum (because a pair of spectra is used in the procedure). The wavenumber ranges used for each kind of measurements in each channel are provided in Table 3. It is important to mention that for the evaluation of the SNR in solar measurements, two spectra recorded during noon time were selected in order to minimize the variability of the solar zenith angle and to use spectra recorded when solar intensity is maximal. For both solar and laboratory open-path spectra, 10 scans recorded with 10 kHz scan speed were

coadded (total integration time 1 min).

**Table 3: Description of the wavenumber region utilized for each channel and for each kind of measurements.**

| Type of measurements | Instrument's channel | Wavenumber range used [cm⁻¹] |
|---|---|---|
| Solar | First | 6515 – 6415 |
| | Second | 4500 – 4400 |
| OP at Laboratory | First | 6200 – 6000 |
| | Second | 4500 – 4300 |

In the Figure 24, the SNR values in the selected spectral regions and both kinds of measurements – open-path and solar – are

presented. For the solar measurements higher scatter of the SNR is found in comparison with the OP laboratory measurements which are more consistent. The SNR values of the solar measurements show a much stronger correlation between the two channels than the SNR values of the open-path measurements (see Figure 25). This higher level of correlation is expected if the variable SNR is due to variable weather conditions. Therefore, the SNR values deduced from the open-path measurements are better suited as an indicator of the SNR performance of each spectrometer. However, even for the laboratory measurements

we expect some artificial variability, as the preamplifier stages are not identical as a consequence different pre-gain and gain settings were used for optimally exploitation of the ADC range. Nevertheless, we can conclude that the SNR typically achieved by the EM27/SUN in a solar spectrum spans the 3 000 to 10 000 cm⁻¹ range, and the SNR of a laboratory open-path spectrum is in the 2 000 to 4 000 cm⁻¹ range for the main channel and 1 000 to 3 000 cm⁻¹ for the CO channel.





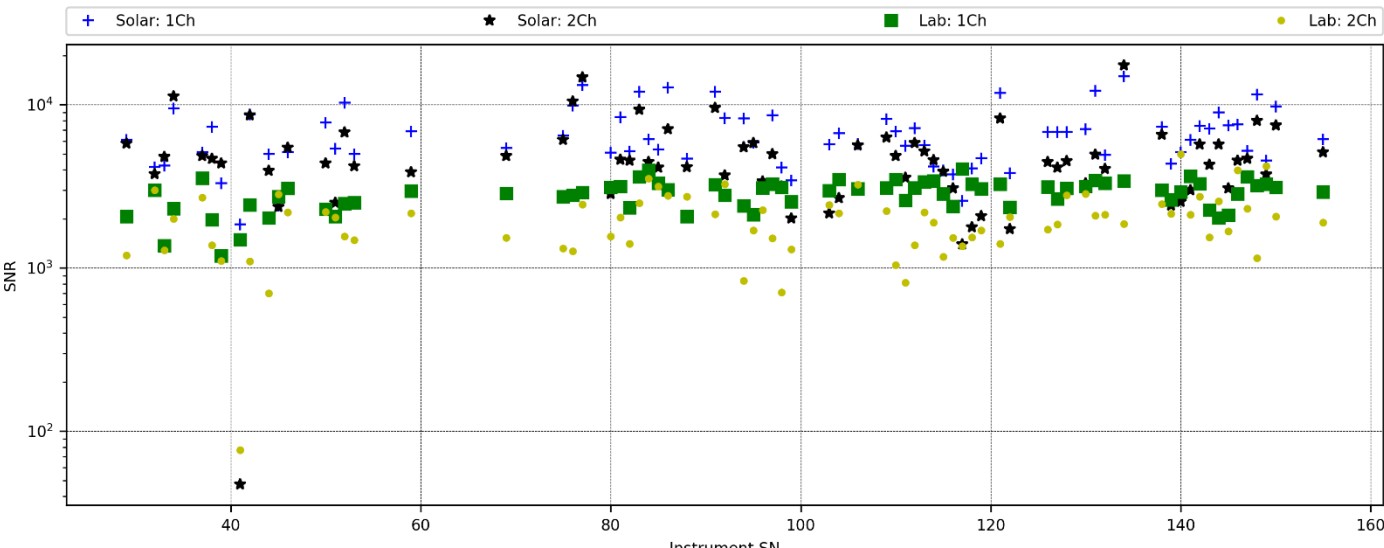

**Figure 24: Instrumental distribution of the SNR for both channels with both kinds of measurements: Solar and OP in the laboratory.**

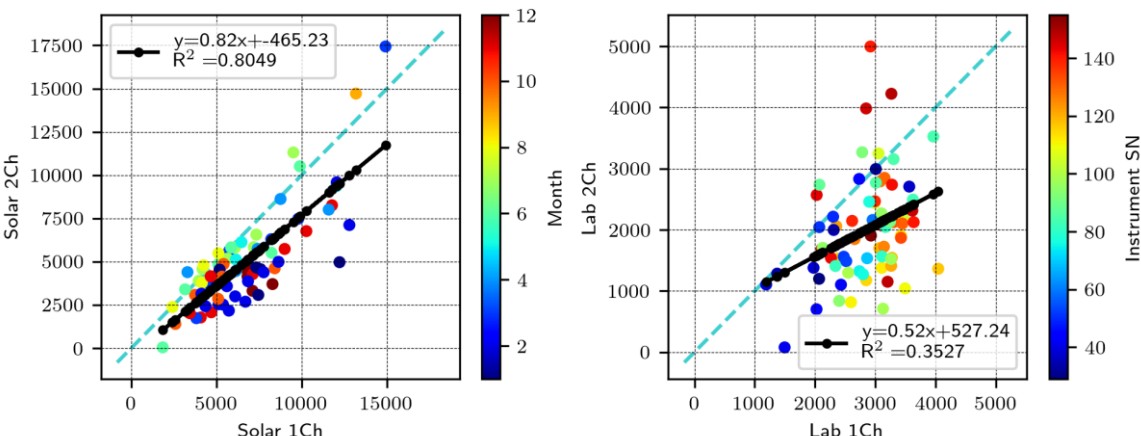

**Figure 25: Correlations of the SNR obtained in channel 1 and 2, for the solar and OP-measurements in the left and right panel of the figure respectively. In the left panel the colour code represents the month of the year when the solar measurements were carried out, for demonstrating the absence of an obvious seasonal signal in the SNR characteristics. In the right panel, the colour code represents the instrument's serial number because these measurements are carried out under controlled laboratory conditions by using a lamp as light source. There might be a slight tendency towards higher SNR in recently built spectrometers.**



## Summary

The analysis of the open-path measurements for deriving the ILS parameters of EM27/SUN FTIR spectrometers were improved and all previous laboratory open-path measurements for the determination of ILS parameters were reanalysed. The revised empirical $H_2O$ line list allows for a significant reduction of fit residuals. The addition of a second spectral window, which can be observed in both channels of the EM27/SUN spectrometer, allows us to identify and quantify significant CO detector misalignments.

In addition to the open-path measurements, a cell filled with $C_2H_2$ was constructed and put into service. The cell measurement can be performed in sequence with the open-path measurement without significant additional effort. It offers similar sensitivity to the ILS parameters, adds redundancy to the calibration process, and the $C_2H_2$ column is expected to be invariant for all EM27/SUN spectrometers. We find an excellent agreement of the retrieved column amount between different spectrometers (1-sigma scatter in the order of 0.01%, see Figure 14 C)). The stability of the COCCON reference spectrometer was investigated and variations of $X_{AIR}$ were found in the 2015 – 2017 period. This variability has a non-negligible impact on the $XCH_4$ calibration results during this period of up to 0.14%. All previous side-by-side measurements reported by Frey et al., 2019, were reanalysed, using the revised ILS parameters and incorporating the correction of the $XCH_4$ calibration factors for the 2015 – 2017 period.

Forty-seven new spectrometers were calibrated before going into operation and several previously investigated spectrometers were recalibrated. The resulting ILS parameters and empirical calibration factors for each target gas are reported. We finally investigated the typical spectral SNR achieved by the EM27/SUN spectrometer in solar and open-path measurements.

We notice a tendency towards improved, more consistent performance of recent EM27/SUN spectrometers. We believe that the continued refinement and continuous application of the quality assurance procedures performed by COCCON in cooperation with the manufacturer of the spectrometers, Bruker, supports this tendency.





## Appendix A

Figure 26: H₂O spectroscopic lines used for this ILS calibration study. The fits (multispectrum fit performed) using HITRAN 2016 and the new empirical COCCON line list are presented in the top and bottom panel respectively. The measured spectra were taken with the IFS125HR at KIT Karlsruhe, at 15° C. The spectral residuals shown are multiplied by 5 in order to be visible.







**Figure 27: Same as Figure 26 but the measured spectra were taken with the IFS125HR at KIT Karlsruhe, at 30° C.**



**Code availability:** Linefit v14.8 used for the ILS characterization can be obtained by contacting Frank Hase (frank.hase@kit.edu). The PROFFAST software, is freely available using the following link: https://www.imk-asf.kit.edu/english/3225.php (last access: 19 September 2021).

**Data availability:** All the data used for this study can be directly requested from the author: Carlos Alberti
(carlos.alberti@kit.edu)

**Team list**

EM27/SUN-partners team: Bianca Baier (Cooperative Institute for Research in Environmental Sciences (CIRES), NOAA Global Monitoring Laboratory), Caroline Bès, Denis Jouglet (Centre National d'Etudes Spatiales (CNES), France), Jianrong Bi (Key Laboratory for Semi-Arid Climate Change of the Ministry of Education, College of Atmospheric Sciences, Lanzhou
University, China), Hartmut Boesch (National Centre for Earth Observation , University of Leicester, UK), André Butz, Ralph Kleinschek (Universität Heidelberg, Germany), Zhaonan Cai (Institute of Atmospheric Physics, Chinese Academy of Sciences, Beijing China), Jia Chen (Technische Universität München, Germany), Sean M. Crowell (University of Oklahoma, USA), Nicholas M. Deutscher, Nicholas B. Jones, David Griffith and Voltaire Velazco (University of Wollongong, Australia), Dragos Ene (National Institute for Research and Development in Optoelectronics (INOE), Romania), Jonathan E. Franklin
(Harvard University, USA), Omaira Garcia (Meteorological State Agency of Spain (AEMET), Spain), Bruno Grouiez, Abdelhamid Hamdouni and Lilian Joly (GSMA group of Molecular and Atmopsheric Spectrometry UMR CNRS, University of Reims, France), Michel Grutter and Wolfgang Stremme (Universidad National Autónoma de México, México), Sander Houweling (Vrije University Amsterdam, The Netherlands), Neil Humpage (National Centre for Earth Observation, University of Leicester, UK), Nicole Jacobs (University of Alaska Fairbanks and Colorado State University), Sujong Jeong ( Seoul
National University (SNU), Korea), Rigel Kivi (Finnish Meteorological Institute (FMI), Finland), Morgan Lopez and Michel Ramonet (Laboratoire des Sciences du Climat et de l'Environment (LSCE), France), Diogo J. Medeiros (Universidade de São Paulo, Brazil), Isamu Morino, Hiroshi Tanimoto and Hirofumi Ohyama (National Institute for Environmental Studies (NIES), Japan), Nasrin Mostafavipak and Debra Wunch (University of Toronto, Canada), Paul I. Palmer (National Centre for Earth Observation, University of Edinburgh, UK), Mahesh Pathakot (National Remote Sensing Centre (NRSC), India), David
Pollard (National Institute of Water and Atmospheric Research Ltd (NIWA), New Zealand), Uwe Raffalski (Swedish Institute of space physics, Kiruna), Robbie Ramsay (NERC Field Spectroscopy Facility, Edinburgh), Mahesh Kumar Sha (Royal Belgian Institute for Space Aeronomy (BIRA-IASB), Brussels, Belgium), Kei Shiomi (Japan Aerospace Exploration Agency, Japan), William Simpson (University of Alaska Fairbanks), Youwen Sun (Anhui Institute of Optics and Fine Mechanics, HFIPS, Chinese Academy of Sciences, Hefei, China), Yao Té (Laboratoire d'Etudes du Rayonnement et de la Matière en
Astrophysique et Atmosphères (LERMA-IPSL), Sorbonne Université, Paris, France), Gizaw Mengistu Tsidu (Botswana International University of Science and Technology), Felix Vogel (Environment and Climate Change Canada, Canada),





Masataka Watanabe (Chuo University, Tokyo Japan), Chong Wei (Shanghai Carbon Data Research Center, Shanghai Advanced Research Institute (SARI), Chinese Academy of Sciences, China), Lu Zhang (National Satellite Meteorological Center (NSMC), China Meteorological Administration, China).


**Author contributions:** CA and FH developed the experimental setup and data analysis procedures and carried out the measurements. CA performed the data analysis and wrote the manuscript with support from FH. FH built and calibrated the cell. MF performed the laboratory measurements from the older instruments and proofread the final article. DD processed the EM27/SUN spectrometer reference unit and the IFS125-LR data set. TB contributed to the calibration effort, proofread and

helped to the improvement the manuscript. AD monitors, steers, and regularly reviews the calibration efforts undertaken for COCCON. GS and RH have contributed to the calibration effort by helping to diagnose and solve any device problem found. JO has continuously supported and encouraged the success of the COCCON project, contributed to the calibration efforts, and to improving the manuscript. The EM27/SUN-partners team collaborated on the calibration efforts and helped to improve the manuscript.


**Competing interests:** The authors declare that they have no conflict of interest.

**Acknowledgement:** We thank M. Kohler for providing the meteorological datasets from IMK-TRO's tower. DE was supported by the Romanian National Core Program (Contract No. 18N/2019). DJM acknowledges the São Paulo Research

Foundation (FAPESP) for financial support (grant 2019/27079-2).

**Financial support:** This study was supported by ESA projects COCCON-PROCEEDS (grant agreement 4000121212/17/I-EF), QA4EO (grant agreement 4000128426/19/NL/FF/ab), and the FRM4GHG 2.0 project, which has received research funding from the European Space Agency's FRM Programme under grant agreement no. 4000136108/21/I-DT-lr


Wollongong TCCON measurements are supported by the Australian Research Council (ARC) grants DP160101598 and LE0668470.

The article processing charges for this open access publication were covered by KIT as a Research Centre of the Helmholtz Association.



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
