# Peer review of "Improved calibration procedures for the EM27/SUN spectrometers of the COllaborative Carbon Column Observing Network (COCCON)"

_Atmospheric Measurement Techniques, 2021_

## Author Comment (AC1)

We thank anonymous referee #1 for evaluating our manuscript and for the very useful comments, which we treat in the following item-by-item. In this author's comment, the points as raised by the reviewer are replicated in blue text, along with the corresponding reply from the authors in black text.

**General comments**

Manuscript continues series of papers devoted to the development of the COCCON network which is based on EM27/SUN FTSs (Fourier transform spectrometer) observations. This type of FTS designed by KIT in close collaboration with Bruker Optics has a number of unique characteristics including portability, robustness, and ease of use. The combination of EM27/SUN FTS together with the state-of-the-art open-source codes (PREPROCESS and FROFFAST) designed at KIT for processing of interferograms and spectra allows provision the highest accuracy/precision values of atmospheric XCO2, XCH4, XH2O and XCO. Since 2014, EM27/SUN spectrometers have been successfully tested in various kinds of environments including a number of intensive field campaigns. The COCCON community is growing rapidly, therefore one of most important tasks is to develop tools to ensure rigorous QA/QC throughout the network. Namely, the manuscript focuses on the improved procedure of EM27/SUN calibration including the analysis of ILS (Instrumental Line Shape) by means of newly designed cell filled with C2H2-air mixture in comparison with standard procedure based on open path measurements of H2O spectral signatures.

The manuscript is well written nevertheless the current version mostly resembles a technical report. In conclusion, it would be useful to give a brief recommendation on how the COCOON community could (or should) implement the findings presented in paper in practice.

We agree that the paper, which reports our recent progress on network calibration work for COCCON shares some resemblance with a technical report. We nevertheless believe that it contains significant new methodological improvements justifying a publication in AMT. Especially, we refine the open-path procedures, introduce an improved $H_2O$ line list for the analysis of open path measurements and the design and commissioning of a $C_2H_2$ reference cell.

We hope the paper will provide a useful reference for both the COCCON community in a narrower sense (instrument operators) as well as for the wider range of COCCON data users. For the first group, the new $H_2O$ line list we created for the analysis and other details of the refined procedure are made available for achieving network-wide improved open-path ILS results (suggestion to use the most accurate pressure reference available, use of the revised internal path length, to check also the CO channel ILS). For the second group, the provided estimate of the performance of all individual spectrometers will be of relevance (then read as a report on the achieved network performance). In the summary, we have added the statement (red):

"We recommend the application of this new refined procedure for characterizing the ILS parameters of the EM27/SUN FTIR spectrometer from open path measurements."

We plan to circulate calibrated $C_2H_2$ cells as an additional handle for recognizing instrumental drifts in the network, this further step is under preparation and will involve all interested COCCON partners. In the current paper, our aim was to demonstrate the feasibility of the new $C_2H_2$ cell method and the quality of results achievable with such a cell. In the summary, we added the statement (red):

"Based on these encouraging results, we plan to circulate $C_2H_2$ cells for demonstrating the level of temporal stability of individual spectrometers and the level of instrument-to-instrument consistency across the network."

We also added further information on the retrieved $C_2H_2$ cell column values from measurements taken with different spectrometers to inform the reader which level of consistency is achieved across different spectrometers (see updated Fig. 16).

**Specific comments**

Thanks for the list of specific comments, which we handle in the following. Please note that we meanwhile discovered that our revised results for spectrometers SN29, SN32, SN50, SN52 and SN53 were suspicious outliers in the resulting instrument specific gas calibration factors when compared to the previous results by Frey et al. (2019). We identified the reason of this problem (we used outdated and not the latest sets of laboratory spectra available for those spectrometers as used by Frey et al. (2019)). We repeated the analysis for these spectrometers and updated accordingly all dependent figures and tables in the paper.

*Abstract, lines 19-20: It is worth mentioning that new calibration cell is filled with air-C2H2 mixture.*

Correct, we use the assumption of some air contamination of the cell (pressure values in table 2), but we did not add deliberately air to the cell content. As the partial pressure of $C_2H_2$ follows from the measured line strengths, this is not a free adjustable parameter if we maintain the assumption that the reported band intensity in HITRAN is correct. For reproducing the line widths measured with the IFS125HR spectrometer, we decided to use the total pressure as tuning parameter. The values (total pressure about 15% higher than partial pressure) might indicate that (1) the reported self-broadening parameters are underestimated (2) the reported band intensity is overestimated (3) the cell actually contains some air contamination – or a combination of all three impacting factors. Following this procedure and line of reasoning, our only reference point is the IFS125HR instrumental line shape, which we assume to equal that of a nominal spectrometer.

We agree to the referee that depending on cell length and band intensity as determined by the chosen target gas, it might indeed be required to work with a significantly diluted target gas (and then this fact should be mentioned), but with the cell, gas, and spectral band used here, "pure" $C_2H_2$ does a reasonable job.

*Line 119: The distance "about 4 m" is mentioned, while Figure 1 caption says about "distanse of 4.20 m". How critical is the precise knowledge of the distance between lamp and the first mirror of EM27/SUN solar tracker?*

Thank for this observation! Before January 2020 this distance was fixed to 4.0 m, after that and because we implemented the cell-measurements together with the open path, we changed that value to 4.20 m. The initial idea of increasing the distance was to preserve the 4 m distance with the cell in the beam and to derive both the cell (20 cm cell length) and open path results from the same measured spectrum. But it turned out that the cell observations require less scans than the open path measurements, so we decided to keep both measurements separated (for excluding any possible residual disturbances on the open path spectra due to the presence of the cell) and as a result continued the measurements with 4.2 m distance. Certainly, this change is reflected in the data analysis of the open path measurements. In the previous calibration paper (Frey et al, 2015) a sensitivity study of retrieved modulation efficiency as function of open path length was carried out and found that the residual variation of the M.E by changing this distance between 3 and 6 meters is only ~ 0.11%, so we do not expect a significant impact of this modification. This residual change detected by Frey et al. (2015) is probably due to spectroscopic issues. Because we revised the $H_2O$ line list, we hoped for a further reduction of this effect and redid the exercise, but the sensitivity remains about the same. We decided to omit these results in the (already lengthy) paper, as the findings by Frey et al. (2015) remain valid.

We have added the following sentence to the manuscript (in red):

"The spectrometer resides on a table, while the lamp is mounted on a tripod at about 4.20 m (4.0 m for instruments calibrated before January 2020) distance from the first mirror of the solar tracker attached to the spectrometer."

Figure 1: In addition to the existing panels it would be helpful providing an extra panel with the side view on the set-up for open-path measurements.

Thanks for the suggestion, we have added the following figure:

[Figure]

Lines 131-132: "The spectrometer is now oriented in such a way that the cell can be conveniently located in the infrared beam on top of the spectrometer housing (see Figure 1 and Figure 4 A).": This is not clear from the photos in Fig.1 and Fig.4A where/how the cell is placed.

Figure 4 was updated with the side-view of the set-up, based on Figure 1, to better show the way how the cell is mounted on the cover of the spectrometer.

[Figure]

Line 138-139: The term "instrument entrance" is not mentioned/presented in the list in Figure 2, so it is unclear which path (inaccessible) is being measured.

The figure has been replaced by the figure below. The instrument's entrance window (we assume that this location is defined by the long-pass filter integrated in the cover of the spectrometer) and the position of the aperture stop inside the spectrometer are now clearly marked to avoid confusion.

[Figure]

**Where:**
- ✓ A') Instrument's entrance
- ✓ A) Long-pass filter
- ✓ B) Reflecting mirrors
- ✓ C) Beam splitter
- ✓ D) Moving mirror
- ✓ E) Aperture ~ 3 mm
- ✓ E') Finely structured optical target
- ✓ E'') Pocket lamp
- ✓ F) Mirror
- ✓ G) Aperture ~ 0.6 mm
- ✓ H) Diffusor
- ✓ I) First channel detector InGaAS
- ✓ J) Professional camera

Fig.2: It would be helpful to indicate the position of an fine optical target in Fig.2.

The figure has been updated in this respect also and now contains the position of the illuminated target (which coincides with the aperture stop of the interferometer).

Lines 193-195: "The latter window resides in the spectral overlap region covered by both detectors, allowing a check for a degraded ILS of the CO channel with respect to the primary channel, because in this spectral window the retrieval of ILS parameters can be performed from both main channel and CO channel spectra." Could authors explain why ILS of the CO channel is degraded?

The ILS of the CO channel is not necessarily degraded. If the alignment of the spectrometer is nominal, then no ILS degradation occurs. In practice, however, the field stop of the primary detector is used as the reference for the interferometric alignment, while the CO detector field stop is adjusted to optically coincide with the field stop of the primary detector (using an additional mirror which decouples some radiation from the primary beam). As a consequence of this procedure, some misalignment between the two detectors might remain and checking the ILS of both detectors is a more stringent procedure for quality assurance than checking only the ILS of the primary detector.

We have added the following statement in the manuscript (in red):

"A dedicated check of the CO channel seems advisable, because the primary channel is used as the reference for the interferometric alignment, while the CO channel is only adjusted to match the alignment of the primary channel."

Line 264: "2.4 Error budget of the cell measurement for measuring ILS parameters of the EM27/SUN spectrometer" More detailed discussion of error budget is expected in section 2.4.

Thanks for the hint, we agree. We have extended the discussion and have included a table for specifically showing the error contribution from spectral noise, cell temperature, and from a correlated disturbance of the reference cell parameters: total pressure ($P_{tot}$) and partial pressure ($P_{part}$). By changing these values by 0.5%. The calibration of the cell parameters turns out to be a critical task (as it contributes a systematic uncertainty to all cell results); we added this information to the text.

| Error source | uncertainty | Propagation on MEA |
|---|---|---|
| Spectral signal-to-noise ratio | 2000 | $1.5 \cdot 10^{-4}$ |
| Temperature | 1 K | $2.5 \cdot 10^{-3}$ |
| Empirical cell pressure parameters (systematic error contribution) | 0.5% | $3.6 \cdot 10^{-3}$ |

Fig.7: The blue dotted line in right panel does not correspond to y=x.

Many thanks for this observation! The Figure legend has been update with the correct equation!

**Technical corrections**

Line 128: An excess colon.

The colon was deleted

Line 129: "In" should be written with a capital letter.

Changed accordingly.

Line 132: An excess parenthesis after " Figure 4 A)".

The Figure 4 is split in A) and B). In that sentence by coincidence, we must have two parenthesis at the end: the referenced Figure part and the closing one.

Line 186: Please clarify the "ca." abbreviation.

We have changed "ca" which is an abbreviation of "circa" for "approximately", which is more conventional than the other one.

Line 505: "Figure 19: ... (derived from high-resolution IFS125-LR spectra using GGG2014, red)...". Is "IFS125-LR" correct here?

Many thanks for pointing this out. In that sentence is not correct, therefore we have changed the ending "LR" by "HR".

---

## Author Comment (AC2)

We would like to thank our anonymous referee#2 for the valuable comments on our manuscript. We have answered point-by-point each of the questions raised by the referee as follows: the original referee's inputs are shown in blue text, while our answer is in black.

The network COCCON focusses on column measurements of greenhouse gases (GHGs). Sub-percent variations are of interest for the GHG columns. Hence the measurements need to resolve these small variations and small biases between different instruments could lead to erroneous conclusions. For this reason the calibration of the instruments is of high importance. The manuscript describes very detailed how the calibrations are performed and compares the new and old calibration methods on the technical level (e.g. modulation efficiency amplitudes). This is certainly very important for documenting the COCCON calibration and also for readers working with FTIR spectrometry. However, currently the paper does not discuss the impact of the improved calibration on the GHG retrievals.

The propagation of ILS errors into the GHG retrievals has been discussed in depth in the PhD thesis of Qiansi Tu, which is available here: https://publikationen.bibliothek.kit.edu/1000095901. We have added this reference to the paper and added in the introduction the statement "The effect of ILS parameter errors on the retrieved column-averaged abundances has been discussed in detail by Q. Tu, 2019 (see chapter 3.4 and figure 3.6 in this work)."

In my opinion it would be good to address the following questions: Does the improved calibration improve the consistency within COCCON and if yes, how much for each gas. I understand that there might be no improvement, because of the instrument specific correction factors. In that case it should be discussed how much the improved calibration procedure influence the instrument specific correction factors.

This is a very interesting question (and a very demanding task). Unfortunately, we do not know the true atmospheric state and the best available reference measurements (provided by the collocated TCCON spectrometer and the COCCON reference) are exploited for the determination of the empirical instrument-specific calibration factors for each gas.

However, there is still a possible argument supporting the assumption that the refined analysis is superior. Ideally, the explicit description of the instrumental deviations from the nominal would remove any discrepancy between the spectrometer under test and the reference spectrometer. In this case, gas-specific calibration factors would not be needed at all (they all would adopt the same value across different spectrometers). This opens up a way to test for progress in the calibration procedures, even if we do not know the true state in the atmosphere. For detecting an improvement, $XCO_2$ is the best candidate. (The quality of the XCO calibration is limited by other factors not treated in this work. A main impact factor on XCO is the presence of weak channelling in the spectra, because CO is a very weak absorber. In case of $XCH_4$, we unfortunately face the drift of the reference spectrometer during the early years (so affecting the $XCH_4$ calibration results by Frey et al. (2019))). The table below provides the scatter of the gas-specific calibration factor for $XCO_2$ following each recipe (assuming ILS as: nominal, the previous Frey et al. (2019) results and by using the results obtained in this study). The numbers indicate that both methods deliver more consistent calibration factors than the "nominal ILS" assumption and the $XCO_2$ test even suggests an advantage of the refined calibration approach.

| Procedure | Empirical STDEV of $XCO_2$ calibration factor between different spectrometers |
| --- | --- |
| Nominal ILS | 9.49839E-4 |
| ILS results from Frey et al, 2019 | 8.56409E-4 |
| ILS results from this work | 7.16057E-4 |

Also it would be good to discuss if these correction factors could be impacted by atmospheric conditions (e.g. variations in water content) of by the environment the instrument is operated in (temperature, …) and if this has been investigated.

With respect to atmospheric impact factors, water content should be of secondary importance. The atmospheric humidity and temperature might affect slightly the GHG retrieval due to various inaccuracies of the model description, which connects the atmospheric state to a modelled spectrum. However, the side-by-side measurements of the spectrometer under test and the reference unit are recorded under identical atmospheric conditions covering the same airmass range, so the small impact of water vapour or temperature should cancel out.

A separate question is whether the parameterisation of the residual mismatch between two spectrometers as a gas-specific calibration factor is adequate. As we do not know the mechanism triggering the discrepancy, this is hard to answer. If the reason of the discrepancy is, e.g., mainly a spectral baseline offset, and then our approach of using a calibration factor might be suboptimal.

From the instrumental viewpoint, observed air mass is the most obvious variable, which might influence the calibration factor (as it affects the amount of radiation accepted by the spectrometer). The figure below shows the $XCO_2$ time series of a side-by-side measurement, shown as function of solar zenith angle SZA. As can be seen, the approach of using a single calibration factor seems to work reasonably well.

[Figure]

*Figure 1: $XCO_2$ of reference and spectrometer under test, as function of SZA, show uncorrected and corrected $XCO_2$ of test spectrometer*

Finally, again from the instrumental viewpoint, we need to address your question whether the EM27/SUN spectrometer shows some sensitivity to local temperature. Possibly the best handle on this is the retrieved XAIR. The reference spectrometer is operated outside and experiences ambient temperatures in the range 0°C to 35°C during the year.

While the PROFFAST Ver. 1 analysis showed a little annual cycle in XAIR values (this is seen in the lowest panel of figure 19), this spurious effect is gone using the latest HITRAN spectroscopy for the retrievals (PROFFAST ver. 2, this new code is currently in beta release). The annual variation of XAIR (if there is any) is within $\pm 0.001$ according to PROFFAST Ver. 2. This is an excellent level of stability,

which does not leave much room for instrumental characteristics depending on temperature. The figure below shows the PROFFAST ver. 2 XAIR results for the whole year 2019 as function of SZA.

[Figure]

*Figure 2: XAir results obtained with Proffast version 2.0 for the reference unit during 2019.*

Technical comment: please note that we meanwhile discovered that our revised results for spectrometers SN29, SN32, SN50, SN52 and SN53 were suspicious outliers in the resulting instrument specific gas calibration factors when compared to the previous results by Frey et al. (2019). We identified the reason of this problem (we used outdated and not the latest sets of laboratory spectra available for those spectrometers as used by Frey et al. (2019)). We repeated the analysis for these spectrometers and updated accordingly all dependent figures and tables in the paper.

---

## Author Response (AR3)

**Reply on RC1: 'Comment on amt-2021-395', Anonymous Referee #1**

We thank anonymous referee #1 for evaluating our manuscript and for the very useful comments, which we treat in the following item-by-item. In this author's comment, the points as raised by the reviewer are replicated in blue text, along with the corresponding reply from the authors in black text.

**General comments**

Manuscript continues series of papers devoted to the development of the COCCON network which is based on EM27/SUN FTSs (Fourier transform spectrometer) observations. This type of FTS designed by KIT in close collaboration with Bruker Optics has a number of unique characteristics including portability, robustness, and ease of use. The combination of EM27/SUN FTS together with the state-of-the-art open-source codes (PREPROCESS and FROFFAST) designed at KIT for processing of interferograms and spectra allows provision the highest accuracy/precision values of atmospheric XCO2, XCH4, XH2O and XCO. Since 2014, EM27/SUN spectrometers have been successfully tested in various kinds of environments including a number of intensive field campaigns. The COCCON community is growing rapidly, therefore one of most important tasks is to develop tools to ensure rigorous QA/QC throughout the network. Namely, the manuscript focuses on the improved procedure of EM27/SUN calibration including the analysis of ILS (Instrumental Line Shape) by means of newly designed cell filled with C2H2-air mixture in comparison with standard procedure based on open path measurements of H2O spectral signatures.

The manuscript is well written nevertheless the current version mostly resembles a technical report. In conclusion, it would be useful to give a brief recommendation on how the COCOON community could (or should) implement the findings presented in paper in practice.

We agree that the paper, which reports our recent progress on network calibration work for COCCON shares some resemblance with a technical report. We nevertheless believe that it contains significant new methodological improvements justifying a publication in AMT. Especially, we refine the open-path procedures, introduce an improved $H_2O$ line list for the analysis of open path measurements and the design and commissioning of a $C_2H_2$ reference cell.

We hope the paper will provide a useful reference for both the COCCON community in a narrower sense (instrument operators) as well as for the wider range of COCCON data users. For the first group, the new $H_2O$ line list we created for the analysis and other details of the refined procedure are made available for achieving network-wide improved open-path ILS results (suggestion to use the most accurate pressure reference available, use of the revised internal path length, to check also the CO channel ILS). For the second group, the provided estimate of the performance of all individual spectrometers will be of relevance (then read as a report on the achieved network performance). In the summary, we have added the statement (red):

"We recommend the application of this new refined procedure for characterizing the ILS parameters of the EM27/SUN FTIR spectrometer from open path measurements."

We plan to circulate calibrated $C_2H_2$ cells as an additional handle for recognizing instrumental drifts in the network, this further step is under preparation and will involve all interested COCCON partners. In the current paper, our aim was to demonstrate the feasibility of the new $C_2H_2$ cell method and the quality of results achievable with such a cell. In the summary, we added the statement (red):

*"Based on these encouraging results, we plan to circulate $C_2H_2$ cells for demonstrating the level of temporal stability of individual spectrometers and the level of instrument-to-instrument consistency across the network."*

We also added further information on the retrieved $C_2H_2$ cell column values from measurements taken with different spectrometers to inform the reader which level of consistency is achieved across different spectrometers (see updated Fig. 16).

**Specific comments**

Thanks for the list of specific comments, which we handle in the following. Please note that we meanwhile discovered that our revised results for spectrometers SN29, SN32, SN50, SN52 and SN53 were suspicious outliers in the resulting instrument specific gas calibration factors when compared to the previous results by Frey et al. (2019). We identified the reason of this problem (we used outdated and not the latest sets of laboratory spectra available for those spectrometers as used by Frey et al. (2019)). We repeated the analysis for these spectrometers and updated accordingly all dependent figures and tables in the paper.

Abstract, lines 19-20: It is worth mentioning that new calibration cell is filled with air-C2H2 mixture.

Correct, we use the assumption of some air contamination of the cell (pressure values in table 2), but we did not add deliberately air to the cell content. As the partial pressure of $C_2H_2$ follows from the measured line strengths, this is not a free adjustable parameter if we maintain the assumption that the reported band intensity in HITRAN is correct. For reproducing the line widths measured with the IFS125HR spectrometer, we decided to use the total pressure as tuning parameter. The values (total pressure about 15% higher than partial pressure) might indicate that (1) the reported self-broadening parameters are underestimated (2) the reported band intensity is overestimated (3) the cell actually contains some air contamination – or a combination of all three impacting factors. Following this procedure and line of reasoning, our only reference point is the IFS125HR instrumental line shape, which we assume to equal that of a nominal spectrometer.

We agree to the referee that depending on cell length and band intensity as determined by the chosen target gas, it might indeed be required to work with a significantly diluted target gas (and then this fact should be mentioned), but with the cell, gas, and spectral band used here, "pure" $C_2H_2$ does a reasonable job.

Line 119: The distance "about 4 m" is mentioned, while Figure 1 caption says about "distanse of 4.20 m". How critical is the precise knowledge of the distance between lamp and the first mirror of EM27/SUN solar tracker?

Thank for this observation! Before January 2020 this distance was fixed to 4.0 m, after that and because we implemented the cell-measurements together with the open path, we changed that value to 4.20 m. The initial idea of increasing the distance was to preserve the 4 m distance with the cell in the beam and to derive both the cell (20 cm cell length) and open path results from the same measured spectrum. But it turned out that the cell observations require less scans than the open path measurements, so we decided to keep both measurements separated (for excluding any possible residual disturbances on the open path spectra due to the presence of the cell) and as a result continued the measurements with 4.2 m distance. Certainly, this change is reflected in the data analysis of the open path measurements. In the previous calibration paper (Frey et al, 2015) a sensitivity study of retrieved modulation efficiency as function of open path length was carried out and found that the residual variation of the M.E by changing this distance between 3 and 6 meters is only ~ 0.11%, so we do not expect a significant impact of this modification. This residual change detected by Frey et al. (2015) is probably due to spectroscopic

issues. Because we revised the $H_2O$ line list, we hoped for a further reduction of this effect and redid the exercise, but the sensitivity remains about the same. We decided to omit these results in the (already lengthy) paper, as the findings by Frey et al. (2015) remain valid.

We have added the following sentence to the manuscript (in red):

"The spectrometer resides on a table, while the lamp is mounted on a tripod at about 4.20 m (4.0 m for instruments calibrated before January 2020) distance from the first mirror of the solar tracker attached to the spectrometer."

Figure 1: In addition to the existing panels it would be helpful providing an extra panel with the side view on the set-up for open-path measurements.

Thanks for the suggestion, we have added the following figure:

[Figure]

Lines 131-132: "The spectrometer is now oriented in such a way that the cell can be conveniently located in the infrared beam on top of the spectrometer housing (see Figure 1 and Figure 4 A).": This is not clear from the photos in Fig.1 and Fig.4A where/how the cell is placed.

Figure 4 was updated with the side-view of the set-up, based on Figure 1, to better show the way how the cell is mounted on the cover of the spectrometer.

[Figure]

Line 138-139: The term "instrument entrance" is not mentioned/presented in the list in Figure 2, so it is unclear which path (inaccessible) is being measured.

The figure has been replaced by the figure below. The instrument's entrance window (we assume that this location is defined by the long-pass filter integrated in the cover of the spectrometer) and the position of the aperture stop inside the spectrometer are now clearly marked to avoid confusion.

[Figure]

Where:
- ✓ A') Instrument's entrance
- ✓ A) Long-pass filter
- ✓ B) Reflecting mirrors
- ✓ C) Beam splitter
- ✓ D) Moving mirror
- ✓ E) Aperture ~ 3 mm
- ✓ E') Finely structured optical target
- ✓ E'') Pocket lamp
- ✓ F) Mirror
- ✓ G) Aperture ~ 0.6 mm
- ✓ H) Diffusor
- ✓ I) First channel detector InGaAS
- ✓ J) Professional camera

Fig.2: It would be helpful to indicate the position of an fine optical target in Fig.2.

The figure has been updated in this respect also and now contains the position of the illuminated target (which coincides with the aperture stop of the interferometer).

Lines 193-195: "The latter window resides in the spectral overlap region covered by both detectors, allowing a check for a degraded ILS of the CO channel with respect to the primary channel, because in this spectral window the retrieval of ILS parameters can be performed from both main channel and CO channel spectra." Could authors explain why ILS of the CO channel is degraded?

The ILS of the CO channel is not necessarily degraded. If the alignment of the spectrometer is nominal, then no ILS degradation occurs. In practice, however, the field stop of the primary detector is used as the reference for the interferometric alignment, while the CO detector field stop is adjusted to optically coincide with the field stop of the primary detector (using an additional mirror which decouples some radiation from the primary beam). As a consequence of this procedure, some misalignment between the two detectors might remain and checking the ILS of both detectors is a more stringent procedure for quality assurance than checking only the ILS of the primary detector.

We have added the following statement in the manuscript (in red):

"A dedicated check of the CO channel seems advisable, because the primary channel is used as the reference for the interferometric alignment, while the CO channel is only adjusted to match the alignment of the primary channel."

Line 264: "2.4 Error budget of the cell measurement for measuring ILS parameters of the EM27/SUN spectrometer" More detailed discussion of error budget is expected in section 2.4.

Thanks for the hint, we agree. We have extended the discussion and have included a table for specifically showing the error contribution from spectral noise, cell temperature, and from a correlated disturbance of the reference cell parameters: total pressure ($P_{tot}$) and partial pressure ($P_{part}$). By changing these values by 0.5%. The calibration of the cell parameters turns out to be a critical task (as it contributes a systematic uncertainty to all cell results); we added this information to the text.

| Error source | uncertainty | Propagation on MEA |
|---|---|---|
| Spectral signal-to-noise ratio | 2000 | $1.5 \cdot 10^{-4}$ |
| Temperature | 1 K | $2.5 \cdot 10^{-3}$ |
| Empirical cell pressure parameters (systematic error contribution) | 0.5% | $3.6 \cdot 10^{-3}$ |

Fig.7: The blue dotted line in right panel does not correspond to y=x.

Many thanks for this observation! The Figure legend has been update with the correct equation!

**Technical corrections**

Line 128: An excess colon.

The colon was deleted

Line 129: "In" should be written with a capital letter.

Changed accordingly.

Line 132: An excess parenthesis after " Figure 4 A)".

The Figure 4 is split in A) and B). In that sentence by coincidence, we must have two parenthesis at the end: the referenced Figure part and the closing one.

Line 186: Please clarify the "ca." abbreviation.

We have changed "ca" which is an abbreviation of "circa" for "approximately", which is more conventional than the other one.

Line 505: "Figure 19: ... (derived from high-resolution IFS125-LR spectra using GGG2014, red)...". Is "IFS125-LR" correct here?

Many thanks for pointing this out. In that sentence is not correct, therefore we have changed the ending "LR" by "HR".

**Reply on RC2: 'Comment on amt-2021-395', Anonymous Referee #2**

We would like to thank our anonymous referee#2 for the valuable comments on our manuscript. We have answered point-by-point each of the questions raised by the referee as follows: the original referee's inputs are shown in blue text, while our answer is in black.

The network COCCON focusses on column measurements of greenhouse gases (GHGs). Sub-percent variations are of interest for the GHG columns. Hence the measurements need to resolve these small variations and small biases between different instruments could lead to erroneous conclusions. For this reason the calibration of the instruments is of high importance. The manuscript describes very detailed how the calibrations are performed and compares the new and old calibration methods on the technical level (e.g. modulation efficiency amplitudes). This is certainly very important for documenting the COCCON calibration and also for readers working with FTIR spectrometry. However, currently the paper does not discuss the impact of the improved calibration on the GHG retrievals.

The propagation of ILS errors into the GHG retrievals has been discussed in depth in the PhD thesis of Qiansi Tu, which is available here: https://publikationen.bibliothek.kit.edu/1000095901. We have added this reference to the paper and added in the introduction the statement "The effect of ILS parameter errors on the retrieved column-averaged abundances has been discussed in detail by Q. Tu, 2019 (see chapter 3.4 and figure 3.6 in this work)."

In my opinion it would be good to address the following questions: Does the improved calibration improve the consistency within COCCON and if yes, how much for each gas. I understand that there might be no improvement, because of the instrument specific correction factors. In that case it should be discussed how much the improved calibration procedure influence the instrument specific correction factors.

This is a very interesting question (and a very demanding task). Unfortunately, we do not know the true atmospheric state and the best available reference measurements (provided by the collocated TCCON spectrometer and the COCCON reference) are exploited for the determination of the empirical instrument-specific calibration factors for each gas.

However, there is still a possible argument supporting the assumption that the refined analysis is superior. Ideally, the explicit description of the instrumental deviations from the nominal would remove any discrepancy between the spectrometer under test and the reference spectrometer. In this case, gas-specific calibration factors would not be needed at all (they all would adopt the same value across different spectrometers). This opens up a way to test for progress in the calibration procedures, even if we do not know the true state in the atmosphere. For detecting an improvement, $XCO_2$ is the best candidate. (The quality of the XCO calibration is limited by other factors not treated in this work. A main impact factor on XCO is the presence of weak channelling in the spectra, because CO is a very weak absorber. In case of $XCH_4$, we unfortunately face the drift of the reference spectrometer during the early years (so affecting the $XCH_4$ calibration results by Frey et al. (2019))). The table below provides the scatter of the gas-specific calibration factor for $XCO_2$ following each recipe (assuming ILS as: nominal, the previous Frey et al. (2019) results and by using the results obtained in this study). The numbers indicate that both methods deliver more consistent calibration factors than the "nominal ILS" assumption and the $XCO_2$ test even suggests an advantage of the refined calibration approach.

| Procedure | Empirical STDEV of $XCO_2$ calibration factor between different spectrometers |
|---|---|
| Nominal ILS | 9.49839E-4 |
| ILS results from Frey et al, 2019 | 8.56409E-4 |
| ILS results from this work | 7.16057E-4 |

Also it would be good to discuss if these correction factors could be impacted by atmospheric conditions (e.g. variations in water content) of by the environment the instrument is operated in (temperature, …) and if this has been investigated.

With respect to atmospheric impact factors, water content should be of secondary importance. The atmospheric humidity and temperature might affect slightly the GHG retrieval due to various inaccuracies of the model description, which connects the atmospheric state to a modelled spectrum. However, the side-by-side measurements of the spectrometer under test and the reference unit are recorded under identical atmospheric conditions covering the same airmass range, so the small impact of water vapour or temperature should cancel out.

A separate question is whether the parameterisation of the residual mismatch between two spectrometers as a gas-specific calibration factor is adequate. As we do not know the mechanism triggering the discrepancy, this is hard to answer. If the reason of the discrepancy is, e.g., mainly a spectral baseline offset, and then our approach of using a calibration factor might be suboptimal.

From the instrumental viewpoint, observed air mass is the most obvious variable, which might influence the calibration factor (as it affects the amount of radiation accepted by the spectrometer). The figure below shows the $XCO_2$ time series of a side-by-side measurement, shown as function of solar zenith angle SZA. As can be seen, the approach of using a single calibration factor seems to work reasonably well.

[Figure]

*Figure 1: $XCO_2$ of reference and spectrometer under test, as function of SZA, show uncorrected and corrected $XCO_2$ of test spectrometer*

Finally, again from the instrumental viewpoint, we need to address your question whether the EM27/SUN spectrometer shows some sensitivity to local temperature. Possibly the best handle on this is the retrieved XAIR. The reference spectrometer is operated outside and experiences ambient temperatures in the range 0°C to 35°C during the year.

While the PROFFAST Ver. 1 analysis showed a little annual cycle in XAIR values (this is seen in the lowest panel of figure 19), this spurious effect is gone using the latest HITRAN spectroscopy for the retrievals (PROFFAST ver. 2, this new code is currently in beta release). The annual variation of XAIR (if there is any) is within $\pm 0.001$ according to PROFFAST Ver. 2. This is an excellent level of stability, which does not leave much room for instrumental characteristics depending on temperature. The figure below shows the PROFFAST ver. 2 XAIR results for the whole year 2019 as function of SZA.

[Figure]

*Figure 2: XAir results obtained with Proffast version 2.0 for the reference unit during 2019.*

Technical comment: please note that we meanwhile discovered that our revised results for spectrometers SN29, SN32, SN50, SN52 and SN53 were suspicious outliers in the resulting instrument specific gas calibration factors when compared to the previous results by Frey et al. (2019). We identified the reason of this problem (we used outdated and not the latest sets of laboratory spectra available for those spectrometers as used by Frey et al. (2019)). We repeated the analysis for these spectrometers and updated accordingly all dependent figures and tables in the paper.

**Reply to Associate Editor minor revision request**

We thank the associate editor for handling our manuscript, the guidance through the review process and for the very useful comments. In this document, the remaining issues raised by the editor are replicated in blue text, along with the corresponding reply from the authors in black text.

Both reviewers have requested that the manuscript be strengthened by making a clearer connection between the calibration methods for instrument line shape that are presented, and the precision and intercomparability of the retrieved CO2 columns. The authors have discussed this in their response, but it should be more clearly included in the manuscript. Specifically:

- In the introduction, add a brief summary of the effect of ILS errors on retrieved column abundances. Currently, the manuscript cites the unpublished work of Q. Tu but does not discuss its results.

Thanks for this observation! We agree that for the motivation of the work presented, a short discussion concerning the propagation of ILS errors would be useful, for that reason, we added the following section in the introduction (in red):

… Stable instrumental characteristics have been demonstrated despite harsh transport and conditions of operation for periods of up to several years (Frey et al., 2015).

The impacts of various error sources on retrieved gas columns and XGAS results have been discussed in detail by Tu, (2019). The error propagation of the ILS uncertainty into retrieved gas columns and XGAS results is dominated by the modulation efficiency amplitude (MEA) parameter, which describes the deviation of the ILS width from the nominal value. We show the resulting error propagation for an assumed $\pm 0.5\%$ error in MEA on the columns and XGAS results in Figure 1 (similar to figure 3.7 presented by Tu, (2019) but covering a somewhat wider SZA range).

As can be seen from the Figure 1, a 0.5% MEA increase introduces an $XCO_2$ decrease in the order of 0.025%, while for $XCH_4$ the opposite behaviour is observed. $XCH_4$ is showing an increase of ~ 0.1% with a very low SZA dependency. In case of XCO, the resulting disturbance shows a zero crossing near 40° SZA and reaches a maximum value of ~ 0.1% change at highest SZA.

[Figure]

**Figure 1: Daily behaviour of the relative difference for dry-air gas and total columns gas amounts, for changes in the original ILS of $\pm 0.5\%$ of the original ILS parameters for COCCON's reference unit SN37 on 31.05.2021 at Karlsruhe, Germany, for Carbon Dioxide, Methane and Carbon Monoxide respectively; and the total column amount for Oxygen.**

In this paper, the open path (OP) method for ILS calculation of EM27/SUN spectrometers as described by (Frey et al., 2015) is significantly improved and applied to further spectrometers…

- Discuss the effect of the improved ILS on XCO2 retrievals. The calculation of the XCO2 precision that is included in the response to Reviewer 2 is a useful approach to quantify this within the paper.

Thanks so much for this suggestion, we have added the section 5.4 at the end of the paper (before the summary and conclusions section), see below (in red):

**5.4 Effects of improved calibration procedures on XCO₂ calibration**

Ideally, the explicit description of the instrumental characteristics from the nominal behaviour as resulting from the calibration procedure would remove any discrepancy between Xgas results derived from different EM27/SUN spectrometers. Instrument-specific calibration factors for each gas as

provided in Table S2 (see the Supplement of this study) would not be required (they would have identical values across different spectrometers). In practice, this is not achievable; instead, the values should be reported and used in the retrieval work. This is due to the fact, that (1) the use of instrument-specific ILS parameters does very likely not cover all kinds of possible instrumental imperfections and (2) the ILS description resulting from the calibration procedure itself has limited accuracy.

This opens up a way to test for verifying progress made in the calibration procedure; as such, progress is expected to make the resulting Xgas calibration factors more uniform across different spectrometers. However, the quality of the XCO calibration is limited by other factors not treated in this work: the main impact factor being weak channelling in the spectra (Blumenstock et al., 2021), because CO is a very weak absorber. In case of $XCH_4$, we unfortunately face the drift of the reference spectrometer calibration during the early years, as discussed in Section 5.1. Therefore, the instrument-specific $XCO_2$ calibration factor appears to be the best available diagnostic. The Table 1 provides the scatter of the gas-specific calibration factor for $XCO_2$ between different spectrometers following three different recipes: (1) assuming nominal ILS parameters, (2) using the ILS parameters of the previous work by Frey et al. (2019) and (3) using the ILS results obtained in this study. The numbers indicate that either method developing instrument-specific ILS parameters delivers more consistent calibration factors than the "nominal ILS" assumption and the refined calibration approach creates the least amount of scatter.

**Table 1: Impact of ILS parameters on XCO$_2$ calibration factors (this statistics encompasses the subset of spectrometers that has been treated in the study by Frey et al., 2019)**

| Procedure | Empirical standard deviation of $XCO_2$ calibration factors between different spectrometers |
|---|---|
| Nominal ILS | 9.49839E-4 |
| ILS results from Frey et al, 2019 | 8.56409E-4 |
| ILS results from this work | 7.16057E-4 |